# Redox regulation of PTPN22 affects the severity of T-cell-dependent autoimmune inflammation

**Jaime James[1], Yifei Chen[2,3], Clara M Hernandez[1†], Florian Forster[1‡], Markus Dagnell[2], Qing Cheng[2], Amir A Saei[4,5], Hassan Gharibi[4], Gonzalo Fernandez Lahore[1], Annika Åstrand[6], Rajneesh Malhotra[7], Bernard Malissen[8], Roman A Zubarev[4,9], Elias SJ Arnér[2,10], Rikard Holmdahl[1,11]***

[1]Division of Medical Inflammation Research, Department of Medical Biochemistry and Biophysics, Karolinska Institutet, Stockholm, Sweden; [2]Division of Biochemistry, Dept. of Medical Biochemistry and Biophysics, Karolinska Institute, Stockholm, Sweden; [3]Department of Gastroenterology, the First Affiliated Hospital of Xi'an Jiaotong University, Shaanxi, China; [4]Division of Physiological Chemistry I, Dept. of Medical Biochemistry and Biophysics Karolinska Institute, Stockholm, Sweden; [5]Department of Cell Biology, Harvard Medical School, Boston, United States; [6]Project Leader Department, Research and Early Development, Respiratory & Immunology, BioPharmaceuticals R&D, AstraZeneca, Gothenburg, Sweden; [7]Translational Science and Experimental Medicine, Research and Early Development Respiratory & Immunology, BioPharmaceuticals R&D, AstraZeneca, Gothenburg, Sweden; [8]Centre d'Immunophénomique, Aix Marseille Université, INSERM, Marseille, France; [9]Department of Pharmacological & Technological Chemistry, I.M. Sechenov First Moscow State Medical University, Moscow, Russian Federation; [10]Department of Selenoprotein Research, National Institute of Oncology, Budapest, Hungary; [11]National and Local Joint Engineering Research Center of Biodiagnosis and Biotherapy, The Second Affiliated Hospital of Xi'an Jiaotong University, Xi'an, China

*For correspondence: Rikard.Holmdahl@ki.se

Present address: †Monash University, School of Public Health and Preventive Medicine, Melbourne, Australia; ‡SCIOTEC Diagnostic Technologies GmbH, Tulln, Austria

**Abstract** Chronic autoimmune diseases are associated with mutations in PTPN22, a modifier of T cell receptor (TCR) signaling. As with all protein tyrosine phosphatases, the activity of PTPN22 is redox regulated, but if or how such regulation can modulate inflammatory pathways in vivo is not known. To determine this, we created a mouse with a cysteine-to-serine mutation at position 129 in PTPN22 (C129S), a residue proposed to alter the redox regulatory properties of PTPN22 by forming a disulfide with the catalytic C227 residue. The C129S mutant mouse showed a stronger T-cell-dependent inflammatory response and development of T-cell-dependent autoimmune arthritis due to enhanced TCR signaling and activation of T cells, an effect neutralized by a mutation in Ncf1, a component of the NOX2 complex. Activity assays with purified proteins suggest that the functional results can be explained by an increased sensitivity to oxidation of the C129S mutated PTPN22 protein. We also observed that the disulfide of native PTPN22 can be directly reduced by the thioredoxin system, while the C129S mutant lacking this disulfide was less amenable to reductive reactivation. In conclusion, we show that PTPN22 functionally interacts with Ncf1 and is regulated by oxidation via the noncatalytic C129 residue and oxidation-prone PTPN22 leads to increased severity in the development of T-cell-dependent autoimmunity.

## Editor's evaluation

This article documents a novel aspect of how T cell activation is regulated by the PTPN22 phosphatase, namely, reversible oxidation, which transiently reduces the activity of PTPN22 to allow the T cell antigen receptor to drive a strong activation signal. This compelling work adds to our understanding of how an immune response is initiated and provides new insights that could be exploited for the development of new drugs to treat immune-mediated diseases.

## Introduction

Complex autoimmune diseases affect 4–5% of the human population, and large efforts have been invested in finding the underlying genetic polymorphisms (*Ye et al., 2018*). Though a major genetic contribution comes from the major histocompatibility complex (MHC) region, many other loci have also been identified. Two important single-nucleotide polymorphisms have emerged, one located in PTPN22 (*Bottini et al., 2004*), a cytoplasmic class I protein tyrosine phosphatase (PTP), and the other in NCF1 (*Olsson et al., 2012*; *Olsson et al., 2017*), a component of the NOX2 complex, controlling induction of reactive oxygen species (ROS) in antigen-presenting cells. Mutations in *Ncf1*, leading to a lower NOX2-dependent ROS response, have been shown to be a major predisposing genetic factor for autoimmune diseases in both mice and humans (*Olsson et al., 2012*; *Olsson et al., 2017*). PTPN22 is primarily a dominant negative regulator of T cell responsiveness, acting by dephosphorylating important target proteins in the T cell signaling machinery, including LCK, FYN, and ZAP70 (*Bottini and Peterson, 2014*; *Stanford and Bottini, 2014*). Studies on PTPN22 have thus far focused on the effects of knocking out the gene or on the autoimmune variant *Ptpn22*[R620W], but with divergent results (*Carmona and Martín, 2018*). Importantly, however, PTPN22 is likely to be redox regulated. Its activity is, as with other PTPs, dependent upon the integrity of a catalytic active site Cys residue. The reactivity of this cysteine makes it particularly susceptible to oxidation via ROS, leading to concomitant abrogation of PTP activity. Reversible oxidation and reduction of such reactive cysteines has previously been shown to be a regulatory mechanism in signal transduction, regulating important T cell downstream signaling molecules such as NF-KB, NRF2, MAPK, and ERK (*Kesarwani et al., 2013*; *Ostman et al., 2011*). Control of redox regulation of PTPs in cells depends upon the balance between inhibitory oxidation of the catalytic Cys residue and its activation by reduction, where the latter is typically maintained by the thioredoxin system. We have now investigated the possibility that redox regulation of PTPN22 could modulate inflammatory pathways in vivo. Interestingly, the PTPN22 catalytic cysteine (C227) has been suggested to form a disulfide bond with another 'back-door' cysteine (C129), possibly altering the threshold for irreversible oxidation of C227 and thereby affecting the redox state of the active site (*Tsai et al., 2009*). By creating a mouse with a cysteine-to-serine mutation at position 129 in PTPN22 (C129S), we could investigate the possible pathophysiological impact of its redox regulatory properties in vivo. We found that mice with this amino acid replacement developed increased T-cell-dependent inflammatory responses due to enhanced T cell receptor (TCR) signaling, which was dependent on NOX2-produced ROS. This correlated well with findings of a lower overall turnover, higher sensitivity to inhibitory oxidation, and a lower capacity of reductive reactivation by the thioredoxin system of recombinant mutant PTPN22[C129S] compared to wild-type PTPN22. These results show that redox regulation of PTPN22 modulates inflammation in vivo, with a lower resistance to oxidation of PTPN22 promoting aggravated disease.

## Results

### Recombinant PTPN22[C129S] has lower catalytic activity, higher sensitivity to inhibition by oxidation, and lower capacity for reductive reactivation by the thioredoxin system

To investigate the potential impact of a C129S replacement in the PTPN22 protein, we recombinantly produced the catalytic domain of the corresponding human wild-type PTPN22 and PTPN22[C129S] mutant proteins (C129 and catalytic C227 residues have the same numbering in both mouse and human). The recombinant proteins were purified to >95% purity as judged by SDS-PAGE, and kinetic parameters were determined using p-NPP as substrate; wild-type PTPN22 enzyme had a basal turnover of 19.6 min$^{-1}$ with a $K_m$ of 4.57 mM while PTPN22[C129S] displayed a turnover of 11.9 min$^{-1}$ and a

$K_m$ of 8.02 mM under the same conditions, showing that the C129S protein has retained PTP activity but with an overall lower catalytic efficiency (*Figure 1A*), which confirms Tsai et al.'s results (*Tsai et al., 2009*).

Next, we wanted to assess the sensitivity of pre-reduced wild-type PTPN22 to inhibition by oxidation. We found that addition of 50 μM $H_2O_2$ to the pure protein led to inhibition of approximately half the activity after 20 min incubation (*Figure 1B*). We also found that addition of bicarbonate together with $H_2O_2$ noticeably potentiated the inactivation (*Figure 1C*), similar to the properties shown earlier for PTP1B that are likely due to the formation of peroxymonocarbonate that reacts more efficiently than $H_2O_2$ with the PTP enzyme (*Dagnell et al., 2019*). When comparing the $H_2O_2$ sensitivity of PTPN22^C129S with that of wild-type PTPN22, we found that the mutant enzyme was clearly more sensitive to inhibition by $H_2O_2$ than wild-type PTPN22, although it had a lower basal turnover (*Figure 1D*). The same effect was also seen in the presence of a functional thioredoxin system of thioredoxin reductase 1 (TrxR1) coupled with thioredoxin (Trx1), but perhaps less so coupled with thioredoxin-related protein of 14 kDa (TRP14) (*Figure 1E*). It should be noted that also PTP1B displays different reactivities with Trx1 and TRP14 (*Dagnell et al., 2021*), which may have physiological importance.

True regulation of PTPN22 by redox mechanisms should require that the oxidized and thereby inactivated enzyme can be subsequently reactivated by reduction as otherwise an oxidizing step could only represent an irreversible off-switch. Since the thioredoxin system has been previously shown to reactivate phosphatases PTP1B and PTEN (*Dagnell et al., 2019*; *Dagnell et al., 2013*; *Schwertassek et al., 2014*), we next tested this property. We found that $H_2O_2$-inactivated PTPN22 species could be reactivated in vitro using either DTT as reductant or the thioredoxin system composed of NADPH together with TrxR1 and Trx1. Interestingly, wild-type PTPN22, but not the PTPN22^C129S mutant, could be directly reactivated by TrxR1 together with NADPH, without inclusion of Trx1 (*Figure 2A*). Subsequent experiments showed that the effect of TrxR1-dependent reactivation was concentration dependent and that the oxidized forms of wild-type PTPN22 were clearly more amenable to direct reactivation by TrxR1 than those of PTPN22^C129S (*Figure 2B*). Since the only difference between PTPN22 and PTPN22^C129S is the integrity of the noncatalytic C129 residue, we reasoned that the direct reductive reactivation by TrxR1+ NADPH of wild-type PTPN22, not seen with the PTPN22^C129S mutant, might indicate that TrxR1 can directly reduce the disulfide involving C129 that may be formed in the wild-type PTPN22 enzyme. To assess this possibility, we subjected both forms of the enzyme to oxidative conditions or to reduction by either DTT or TrxR1, and then analyzed the enzymes on a nonreducing SDS-PAGE. Indeed, only PTPN22 but not PTPN22^C129S displayed a second faster migrating form of the protein upon oxidation that disappeared upon reduction with DTT or TrxR1 (*Figure 2C*). The effect was seen with several concentrations of $H_2O_2$, and DTT treatment could always revert the double band of PTPN22 (*Figure 2—figure supplement 1*). We also found that the reactivation of wild-type PTPN22, with either DTT or the thioredoxin system (TrxR1 alone, or coupled with either Trx1 or TRP14), was always more efficient than that of the mutant PTPN22^C129S enzyme, also in cases where prior oxidation was further potentiated by bicarbonate (*Figure 2—figure supplement 2*).

These findings with recombinant human PTPN22 and PTPN22^C129S enzymes suggest that the mutant is more amenable to oxidation and that the thioredoxin system is less efficient in reactivating its oxidized form. The notion that TrxR1 directly can reduce the disulfide, which is not formed in the mutant enzyme, but which may be formed upon oxidation of wild-type PTPN22, was interesting, as to our knowledge no PTP has earlier been shown to form a disulfide species that is a direct substrate of TrxR1. Based on these findings, we propose a model for redox regulation of PTPN22 (*Figure 2D*), which illustrates how PTPN22^C129S can be more easily inactivated than the wild-type enzyme. Next, we wished to assess if mutant PTPN22^C129S yields any phenotypic effects in mouse models of inflammation.

## Establishment of PTPN22^C129S mutant mice

To study redox-dependent regulation of PTPN22 in vivo and its possible downstream effects on inflammation, we generated a mouse with the PTPN22^C129S mutation (schematic in *Figure 3A*). This would disrupt the proposed mechanism by which the catalytic C227 can be protected through formation of a disulfide that can be reduced by TrxR1 (*Figure 2D*), thus making PTPN22^C129S in the mice more prone to inactivation by oxidation. The mice were backcrossed to C57Bl/6N mice together with a MHC region with the H-2q haplotype, making it susceptible to autoimmune arthritis (*Brunsberg et al., 1994*), and also with the *m1j* mutation of *Ncf1* (*Hultqvist et al., 2004*), allowing interaction studies.

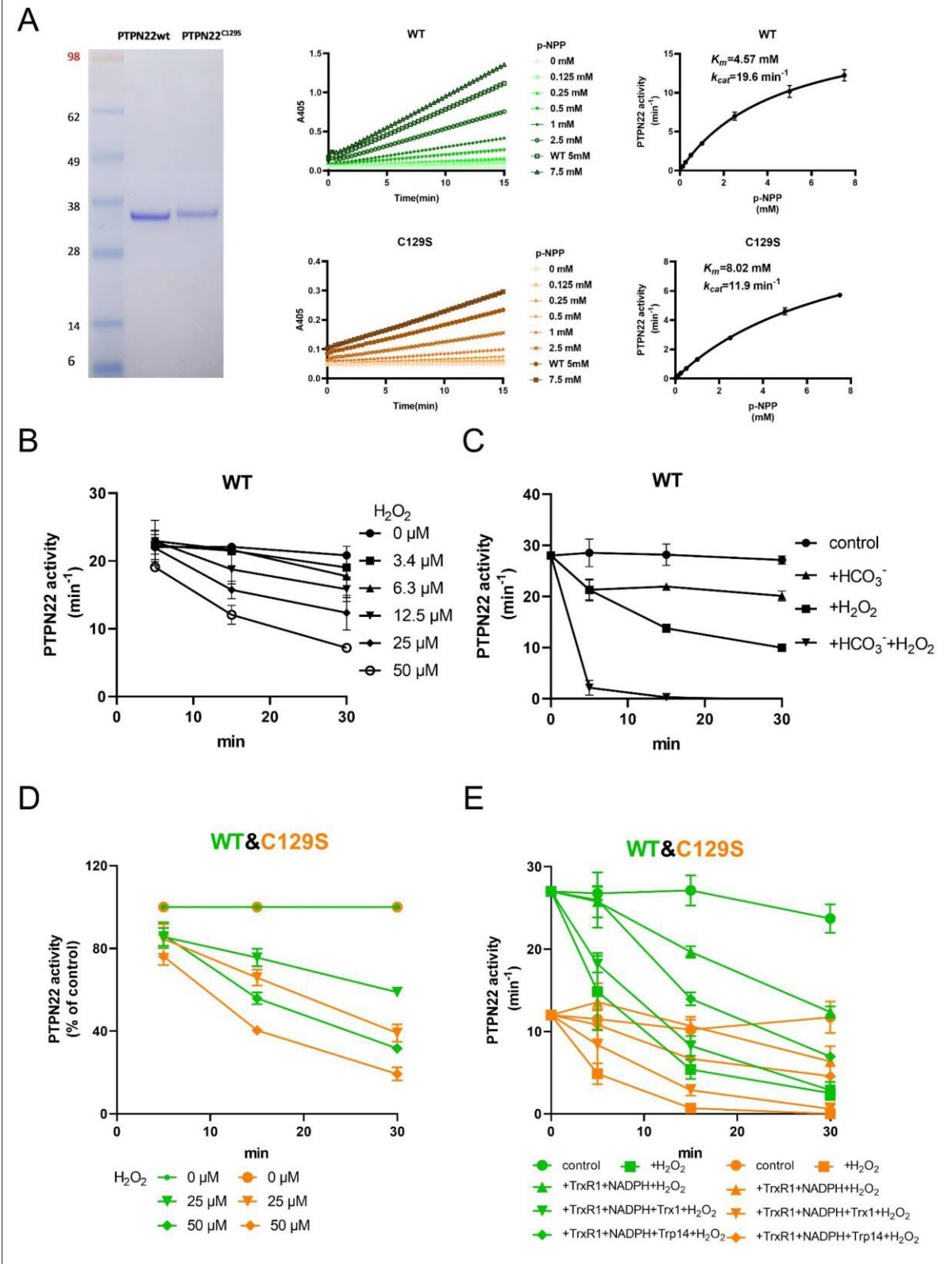

**Figure 1.** PTPN22$^{C129S}$ is more prone to oxidative inactivation. (**A**) Purified human wild-type and PTPN22$^{C129S}$ proteins are shown as analyzed on SDS-PAGE, and kinetic parameters were measured using p-NPP as substrate as shown to the right. Data points represent mean ± SD (error bars) (n = 3). (**B**) PTPN22 (1400 nM) was treated with indicated concentrations of $H_2O_2$ and then assayed for PTP activity at the indicated times. PTPN22 activity is given in min$^{-1}$ (mol of product/mol of enzyme/min). Data points represent mean ± SD (error bars) (n = 3). (**C**) PTPN22 (1400 nM) activity was measured

*Figure 1 continued on next page*

*Figure 1 continued*

as in (**A**) but with buffer control (●), 25 mM bicarbonate (▲), 30 μM $H_2O_2$ (■), or the 896 combination of 30 μM $H_2O_2$ and 25 mM bicarbonate (▼). Data points represent mean ± SD (error bars) (n = 4). (**D**) 897 Wild-type (green) and PTPN22$^{C129S}$ (orange) proteins were treated with indicated $H_2O_2$ concentrations and then assayed for PTP activity. Data points represent mean ± SD (error bars) (n = 3), and activities are given as percentage of buffer control for each enzyme. (**E**) Pre-reduced wild-type (green) and PTPN22$^{C129S}$ (orange) proteins were first preincubated for 10 min either in only buffer (■), with 300 μM NADPH and 0.25 μM TrxR1 (▲), with 300 μM NADPH, 0.25 μM TrxR1 and 10 μM Trx1(▼), or with 300 μM NADPH, 0.25 μM TrxR1, and 10 μM Trp14 (◆), whereupon $H_2O_2$ (100 μM) and 1 mM $NaN_3$ were added, except to the control (●), and samples were taken at indicated times for measurement of PTP activity. Data points represent mean ± SD (error bars) (n = 3).

The online version of this article includes the following source data for figure 1:

**Source data 1.** Raw data for catalytic activity of wild-type and mutant PTPN22.

**Source data 2.** Raw data for PTPN22 activity after $H_2O_2$ treatment.

**Source data 3.** Raw data for PTPN22 activity after $H_2O_2$ treatment with and without bicarbonate.

**Source data 4.** Raw data for comparison of PTPN22 activity between wild-type and mutant after $H_2O_2$ treatment.

**Source data 5.** Raw data for comparison of PTPN22 activity between wild-type and mutant after $H_2O_2$ treatment in the presence of reducing systems.

**Source data 6.** Uncropped gel image showing wild-type and mutant PTPN22.

## PTPN22$^{C129S}$ enhances Th1-mediated inflammation via NOX2-mediated ROS

To study the effect of PTPN22$^{C129S}$ on cell-mediated immunity, we used the delayed-type hypersensitivity (DTH) model, which is known to drive inflammation via IFNγ-producing type 1 T helper cells (Th1) (*Allen, 2013*). Littermate wild-type and PTPN22$^{C129S}$ mice were immunized with heterologous collagen type II (Col2), challenged on day 8 by injection with the same antigen in the ear and subsequently swelling was assessed. We observed increased ear pinna thickness in PTPN22$^{C129S}$ mice 24, 48, and 72 hr after challenge compared to littermate wild-type mice (*Figure 3B*). In contrast, PTPN22$^{C129S}$ did not enhance ear swelling in NCF1 mutant mice with a lack of NOX2 activity (*Figure 3C*, *Olofsson et al., 2003*). To assess the antigen-specific T cell response, we restimulated cells from inguinal lymph nodes with the immunodominant COL2 peptide (259–273) and observed a marked increase in IFNγ and IL-2-producing PTPN22$^{C129S}$ cells when comparing wild-type supporting a Th1 phenotype (*Figure 3D and E*). No difference was observed in the NCF1 mutant background. Within the ears, flow cytometry analysis showed that the challenge with COL2 increased percentages of CD45+ leucocytes and CD45+ TCRb+ cells compared to PBS injection with the NCF1 mutants showing higher cell infiltration even in PBS-injected ears (*Figure 3F*). Additionally, we observed higher CD4 expression in COL2-injected PTPN22 ears compared to wild-type with no difference in the NCF1 mutants (*Figure 3G*). To further support the Th1 phenotype, we performed qPCR analysis of the ears that showed increased expression of CXCR3 in inflamed PTPN22$^{C129S}$ ears (*Figure 3H*).

In the periphery, immunized PTPN22$^{C129S}$ mice had increased cell numbers in the spleen compared to wild-type (*Figure 3I*). Within the inguinal lymph nodes, we observed a reduction in CD4+ T cells, which expressed higher levels of CD44, a marker for effector-memory T cells (*Figure 3J*). This was not seen in circulating T cells within the blood; however, there was a significant reduction in FOXP3+ T cells in PTPN22$^{C129S}$ mice 48 hr after initial immunization (*Figure 3—figure supplement 1*).

As we observed differences in T cell activation levels, we wanted to address the role of PTPN22$^{C129S}$ in antigen-specific T cells. To do this, we used the Vβ12-transgenic mouse (Vβ12-tg), which expresses a TCRβ chain specific for the galactosylated COL2 (260–270) epitope and can be tracked using the clonotypic B22a1 antibody. Upon priming of Vβ12-tg mice with COL2, these T cells expand 50-fold, acquire an activated phenotype, and play an important role in the early phase of the arthritogenic immune response (*Merky et al., 2010*). Upon immunization with COL2, PTPN22$^{C129S}$ did not affect the expansion of antigen-specific Vβ12+ T cells per se (*Figure 3K*), but did change their activation status: COL2-reactive T cells in the periphery expressed higher levels of CD44 on CD4+ T cells compared to wild-type (*Figure 3K*). In the DTH model, we could also observe higher B22a1 expression among CD4+ T cells in the inflamed ears of Vβ12.PTPN22$^{C129S}$ mice (*Figure 3L*).

Together, these results indicate that the oxidation-prone PTPN22$^{C129S}$ mutant enhances T-cell-mediated inflammation. Conversely, wild-type PTPN22 with a higher basal activity and more resistance to oxidation counteracts these inflammatory processes.

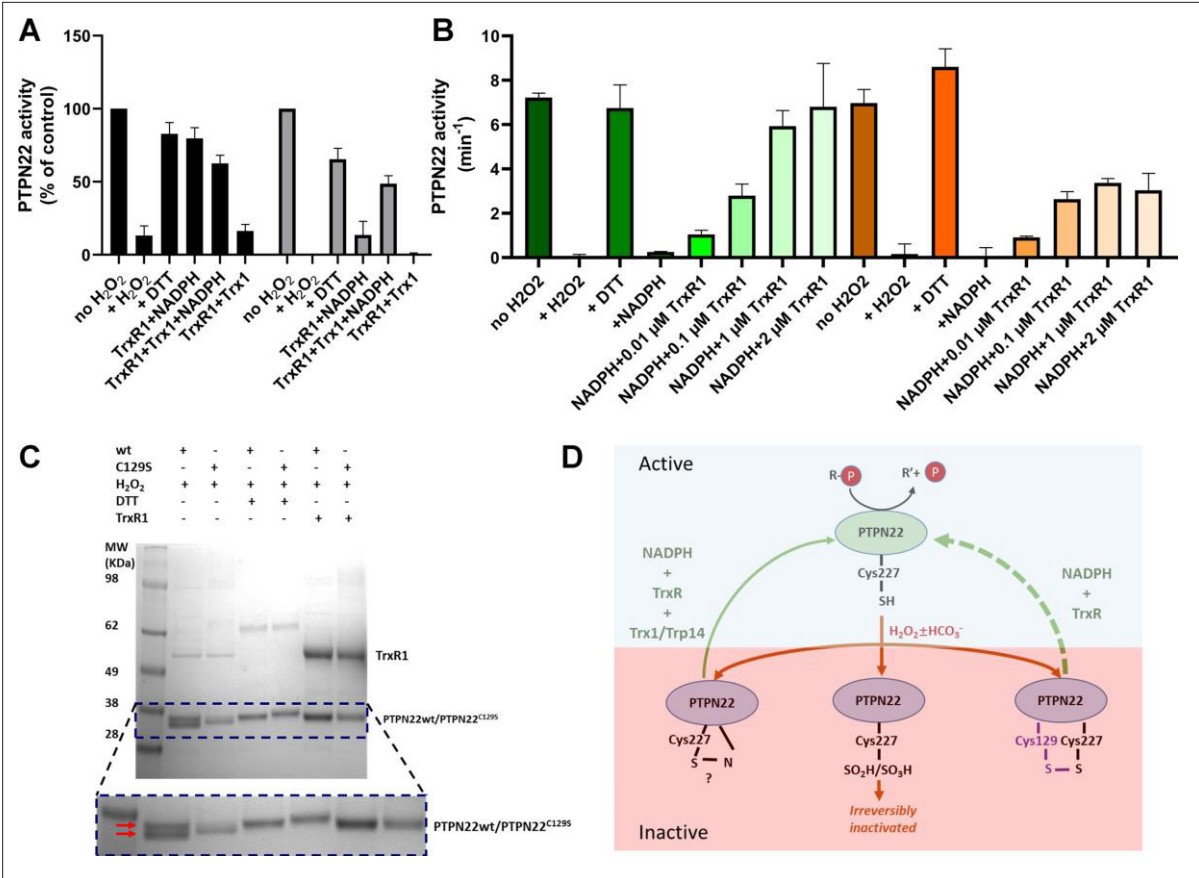

**Figure 2.** PTPN22 can be reactivated by TrxR1. (**A**) Pre-reduced wild-type (black) or PTPN22$^{C129S}$ (gray) proteins were treated with $H_2O_2$ (1 mM) for 5 min whereupon residual $H_2O_2$ was removed by addition of catalase. Thereafter, the enzymes were incubated for 60 min at 37°C with either only buffer, DTT (10 mM), or with combinations of TrxR1 (2.5 µM) and NADPH (300 µM) and Trx1 (10 µM), as indicated. Finally, the samples were analyzed for PTP activity, with activity reported as percentage of control samples treated identically but without addition of $H_2O_2$. (**B**) Pre-reduced PTPN22 (green) and PTPN22$^{C129S}$ (orange) were treated as described in (**A**), reactivated with either 10 mM DTT, or NADPH (300 µM) and different concentrations of TrxR1 as indicated. After 60 min at 37°C, samples were analyzed for PTP activity, given as turnover in min$^{-1}$ (mol of product/mol of enzyme/min). (**C**) SDS-PAGE analysis of PTPN22 or PTPN22$^{C129S}$ treated similarly to the assay shown in (**A**) but incubated with $H_2O_2$ (100 µM), DTT (10 mM), or TrxR1 (0.5 µM together with 300 µM NADPH), as indicated in the figure. The samples were subsequently analyzed on a nonreducing SDS-PAGE. Note the presence of a double band only in oxidized PTPN22, but not in oxidized PTPN22$^{C129S}$, which disappears upon incubation with either DTT or TrxR1 (double red arrows in the enlarged portion of the gel picture). (**D**) A proposed model for redox regulation of PTPN22. Only the reduced form of the enzyme is active (top, light green), which upon oxidation with $H_2O_2$ and as further facilitated by bicarbonate ($HCO_3$.) can form several oxidized species (bottom, purple), possibly being either a sulfenyl amide as suggested for PTP1B or some other species that require the complete thioredoxin system for reactivation (left), as well as irreversibly oxidized with the catalytic C227 converted to sulfinic/sulfonic acid (middle), or by forming a disulfide with C129 (right). This disulfide was here found to be amenable to reduction directly by TrxR1, but as the PTPN22$^{C129S}$ mutant cannot form this species, it can also not be reactivated by that reduction path (dashed green arrow). Both PTPN22 and PTPN22$^{C129S}$ can, however, be reactivated by the thioredoxin system from its other nonirreversibly oxidized states (solid green arrow).

The online version of this article includes the following source data and figure supplement(s) for figure 2:

**Source data 1.** Raw data for reactivation of wild-type and mutant PTPN22.

**Source data 2.** Raw data for reactivation of wild-type and mutant PTPN22 with varying TrxR1 concentrations.

**Source data 3.** Uncropped SDS-PAGE analysis of PTPN22 treated with $H_2O_2$ and then reactivated with DTT or TrxR1.

**Figure supplement 1.** Reproducibility and $H_2O_2$ concentration dependence of the double band appearing in oxidized PTPN22 but not in PTPN22$^{C129S}$, always fully reducible using subsequent treatment with DTT.

**Figure supplement 1—source data 1.** Unedited image showing the presence of a double band in oxidized PTPN22, but not in PTPN22 C129S.

**Figure supplement 2.** C129S mutant PTPN22 is less sensitive to TrxR1 reactivation compared to WT PTPN22.

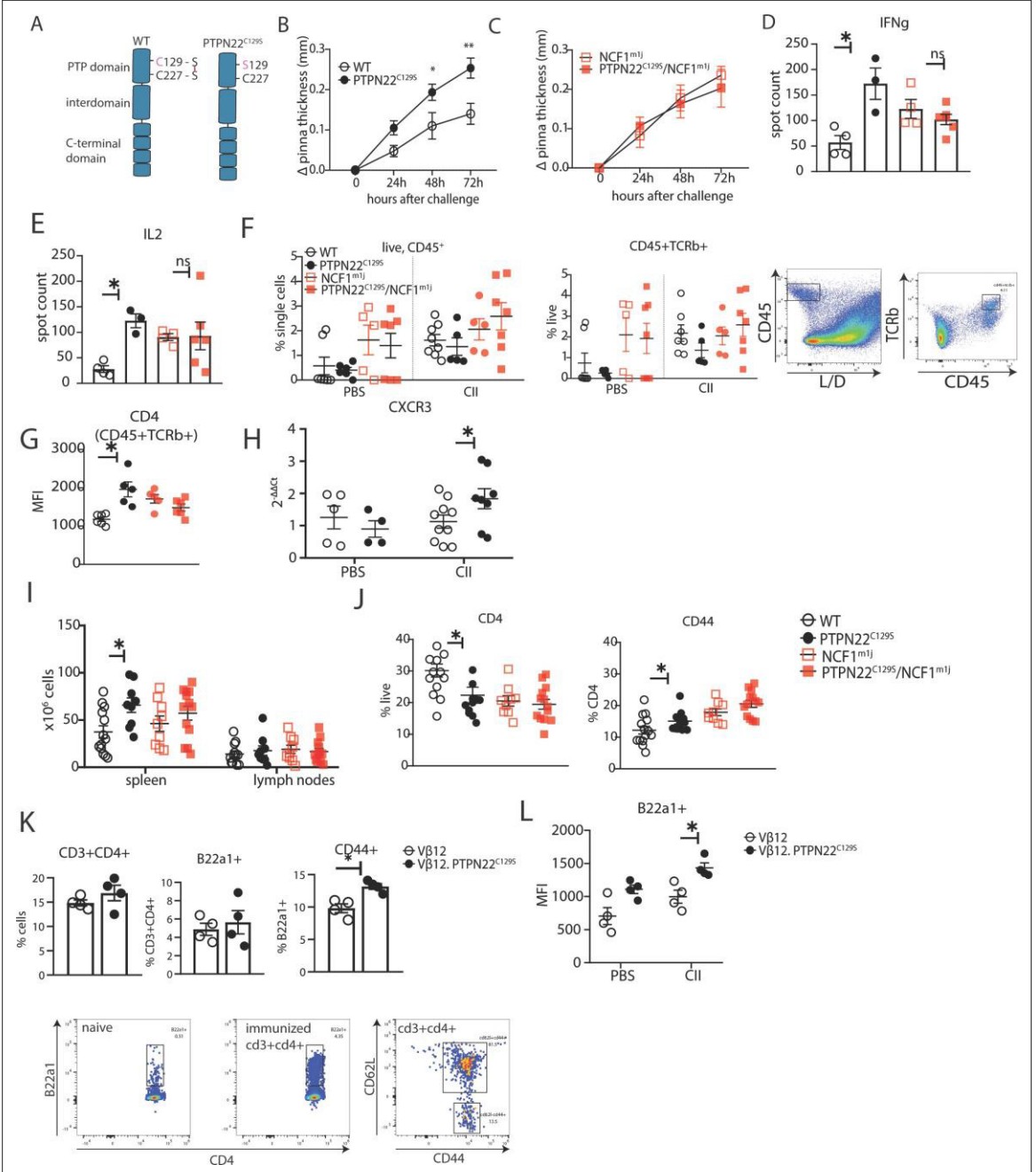

**Figure 3.** PTPN22[C129S] enhances Th1-mediated inflammation via NOX2-mediated reactive oxygen species (ROS). (**A**) Schematic representation of wild-type and PTPN22[C129S] proteins; in the wild-type protein, the catalytic cysteine 227 may form a disulfide bond with cysteine 129, whilst in the mutant, cysteine 129 is mutated into a serine, abrogating this disulfide bond. (**B–L**) Mice were immunized according to delayed-type hypersensitivity (DTH) protocol with collagen type II (Col2), and cell populations/recall responses were analyzed 72 hr after challenge. (**B, C**) Ear swelling of littermate wild-type (n = 8), PTPN22[C129S] (n = 6), NCF1[m1j] (n = 5) and PTPN22[C129S]/NCF1[m1j] (n = 7) mice at indicated time points after Col2 challenge. Swelling shown as mean ± SEM (two-way ANOVA). Data is representative of four experiments. (**D, E**) Measurement of antigen-specific interferon-γ (IFNγ)/IL-2 T cell response against the immunodominant T cell epitope (rCol22259-273) of Col2 as measured by ELISpot in inguinal lymph nodes. (**F**) Frequencies of indicated cell subsets in PBS-injected and Col2-injected ears measured by flow cytometry; representative gating shown. (**G**) Mean fluorescence intensity (MFI) of CD4 within Col2-injected ears measured by flow cytometry. (**H**) qPCR analysis of CXCR3 expression in PBS and Col2-injected ears. (**I**) Cell count in peripheral lymphoid organs at termination of DTH protocol. (**J**) Frequencies of indicated cell subsets within inguinal lymph nodes measured by flow cytometry. (**K**) Frequencies of indicated subsets in inguinal lymph nodes of Vβ12/Vβ12.PTPN22[C129S] mice; flow cytometry plots show expansion of B22a1+ T cells upon immunization and representative gating for CD62L and CD44. (**L**) MFI of B22a1 in PBS/Col2-injected ears. Error bars represent mean ± SEM.

The online version of this article includes the following source data and figure supplement(s) for figure 3:

*Figure 3 continued on next page*

*Figure 3 continued*

**Source data 1.** Raw data for ear pinna thickness in the delayed-type hypersensitivity model.

**Figure supplement 1.** Flow cytometry measurement of indicated cell subsets in blood in mice 0, 48, and 96 hr after primary immunization with Col2.

**Figure supplement 2.** Clinical score (mean ± SEM), incidence, and serum antibody levels at termination of collagen-induced arthritis in littermate mice.

## PTPN22$^{C129S}$ enhances T cell responses and development of arthritis

As PTPN22 is heavily associated with the development of autoimmunity, we also sought to explore the effects of PTPN22$^{C129S}$ on arthritis development by using the glucose-6-phosphate-isomerase peptide (GPIp)-induced arthritis model (GIA), which causes acute autoimmunity that resembles the early stages of rheumatoid arthritis, and which is regulated by NOX2-derived ROS (*Pizzolla et al., 2013*; *Pizzolla et al., 2012*; *Pizzolla et al., 2012*). As shown in *Figure 4A*, we observed increased arthritis severity in PTPN22$^{C129S}$ mice from day 10 up to day 22 after onset as well as increased incidence in mutant mice. On the NCF1-deficient background, we observed a trend in the opposite direction with PTPN22$^{C129S}$ mice showing less disease and no significant difference in disease incidence (*Figure 4B*).

Increased disease was further supported by higher antigen-specific T cell responses in PTPN22$^{C129S}$ mice as shown by increased numbers of IFNγ-secreting T cells, both upon GPIp and anti-CD3/anti-CD28 stimulation (*Figure 4C*). CD4 T cells in the draining lymph nodes exhibited increased activation

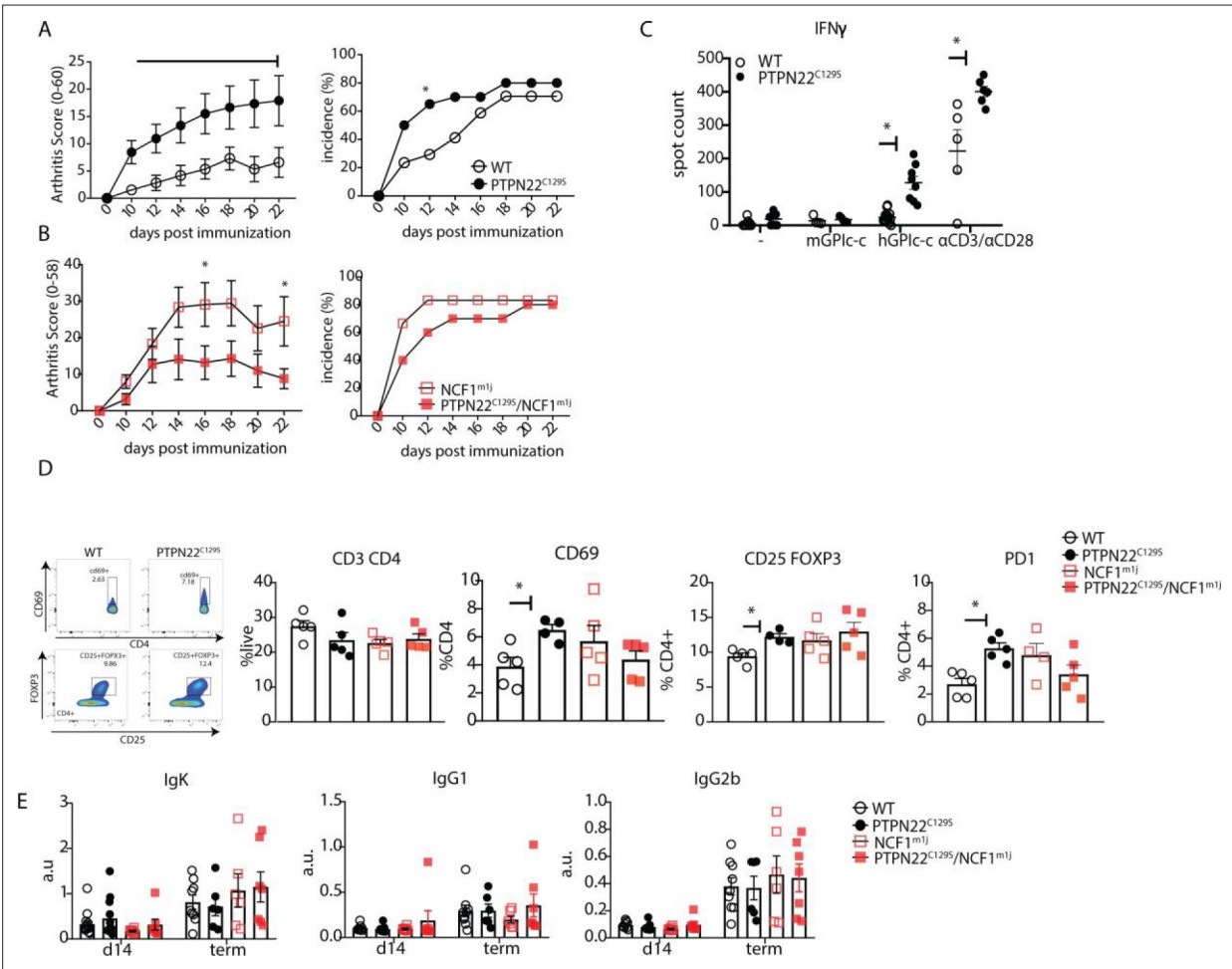

**Figure 4.** PTPN22$^{C129S}$ enhances arthritis development. Arthritis was induced using human glucose-6-phosphate isomerase peptide (GPIp) and inflammation was monitored over 22 days. (A, B) Clinical score (mean ± SEM) and incidence of littermate WT (n = 17), PTPN22$^{C129S}$ (n = 20), NCF1$^{m1J}$ (n = 12), and PTPN22$^{C129S}$/NCF1$^{m1J}$ (n = 10) mice immunized with GPIp. Data from two pooled experiments (two-way ANOVA). (C) Antigen-specific interferon-γ (IFNγ) T cell response against mouse/human GPIp as well as anti-CD3/anti-CD28 in lymph nodes by ELISpot. (D) Percentages of indicated cell subsets in lymph nodes at measured by flow cytometry; representative gating for CD69 and CD25+ FOXP3+ cells shown. (E) Levels of serum antibodies against hGPI day 14 post immunization and at arthritis endpoint measured by ELISA. Error bars represent mean ± SEM.

as evidenced by higher numbers of CD4+ CD69+, CD4+ CD25+ FOXP3+ and CD4+ PD1+ T cells (*Figure 4D*). No differences were observed in serum antibody levels against the hGPI peptide (*Figure 4E*). In summary, PTPN22$^{C129S}$ enhances autoimmune inflammation due to enhanced T cell reactivity.

## PTPN22$^{C129S}$ enhances T cell signaling in vitro and slightly affects thymic T cell development

To understand where the observed T cell activation phenotype in PTPN22$^{C129S}$ mice originates from, we first confirmed that PTPN22 expression was comparable between wild-type, PTPN22$^{C129S}$, NCF1$^{m1J}$, and PTPN22$^{C129S}$/NCF1$^{m1J}$ mice both via mRNA and protein analysis using splenocytes or sorted CD4+ T cells, respectively (*Figure 5A and B*, *Figure 5—figure supplement 1*). Next, we analyzed expansion of naïve immune cell populations. PTPN22$^{C129S}$ was found to not affect immune cell populations in peripheral lymphoid organs in naïve mice; the percentages of B cells (B220+), T cells (CD3+), dendritic cells (CD11c+), and macrophages (CD11B+) were comparable to wild-type littermates as was MHC II expression on antigen-presenting cells (*Figure 5—figure supplement 2*).

As PTPN22 has been previously shown to affect T cell tolerance (*Maine et al., 2012*; *Salmond et al., 2014*), we analyzed T cell development in the thymus. We observed a slight reduction in the percentage of CD4-CD8- (DN) population whilst CD4+CD8+ (DP), CD4+, and CD8+ were comparable between groups. Further fractionation of the early CD4-CD8- progenitor population showed a slight increase in DN1 (CD44+CD25-) and decrease in DN4 (CD44-CD25-) populations in PTPN22$^{C129S}$ thymi (*Figure 5C*). No changes were observed in TCRb, CD5, and CD69 expression (*Figure 5C*). Percentages of CD4+ and CD8+ T cells as well as expression of CD44, and CD69 activation markers on CD4+ T cells were not affected (*Figure 5D*). Furthermore, percentages of CD4+ FOXP3+ and CD4+ FOXP3+ CD25+ cells were comparable between groups in thymus and peripheral lymphoid organs (*Figure 5D*).

As we consistently observed a differential T cell phenotype in mice with mutated PTPN22$^{C129S}$ in vivo, we studied T cell function in vitro. To assess signaling, we measured Ca$^{2+}$-flux, one of the earliest signaling events upon TCR engagement and observed slightly increased intracellular Ca$^{2+}$ in PTPN22$^{C129S}$ CD4 T cells upon anti-CD3 stimulation (*Figure 5E*). Proliferation of CD4 T cells, as assessed by CellTrace dilution, was also increased upon anti-CD3/anti-CD28 stimulation (*Figure 5F*). We observed increased activation of ex vivo-stimulated CD4 T cells as measured by the early T cell activation marker CD69 as well as markedly different IFNɣ production resembling the in vivo results (*Figure 5G and H*).

## Enhanced phosphorylation of PTPN22 targets and enhanced PKC expression in PTPN22$^{C129S}$ cells

To assess whether the differences in T cell reactivity could be correlated with altered PTPN22 catalytic activity, we analyzed the phosphorylation of its targets Fyn, LCK, and Zap70 using lymph node cells. TCR stimulation of wild-type and mutant cells revealed increases in phosphorylation of tyrosines Y420 and Y394 of the Src family kinases Fyn and LCK, respectively, as well as of Y493 of Zap70, thus agreeing well with lower intracellular phosphatase activity of the PTPN22$^{C129S}$ mutant protein (*Figure 6A and B*) and confirming findings with recombinant PTPN22 proteins.

Next, we wanted to assess if downstream T cell-signaling mediators were affected by PTPN22$^{C129S}$. TCR signaling induces activation of protein kinase C (PKC)-θ, an essential player in peripheral T cell activation. Compared to wild-type, PTPN22$^{C129S}$ lymph node cells showed constitutive phosphorylation of Y538 on PKC-θ, independent of TCR stimulation, whereas in wild-type cells phosphorylation increased with TCR stimulation, hinting at higher basal activity of PKC-θ in the mutant (*Figure 6C*). To assess how changes in the redox status affect T cell signaling, we used buthionine sulphoximine (BSO), an inhibitor of ɣ-glutamyl-cysteine synthetase, which is essential for glutathione synthesis. Treatment with BSO modulates the redox systems in cells by lowering the GSH levels and without directly affecting the thioredoxin system, at least initially before compensatory mechanisms may be activated. Here, it initially reverted the phospho-PKC-θ levels in the mutant to be comparable with wild-type, followed by a slower TCR-activation-dependent phosphorylation (*Figure 6C*). Next, we investigated how differential signaling was affecting the proteome signature using sorted CD4+ T cells from wild-type and PTPN22$^{C129S}$ mice. Mass spectrometric proteomics analysis of activated cells

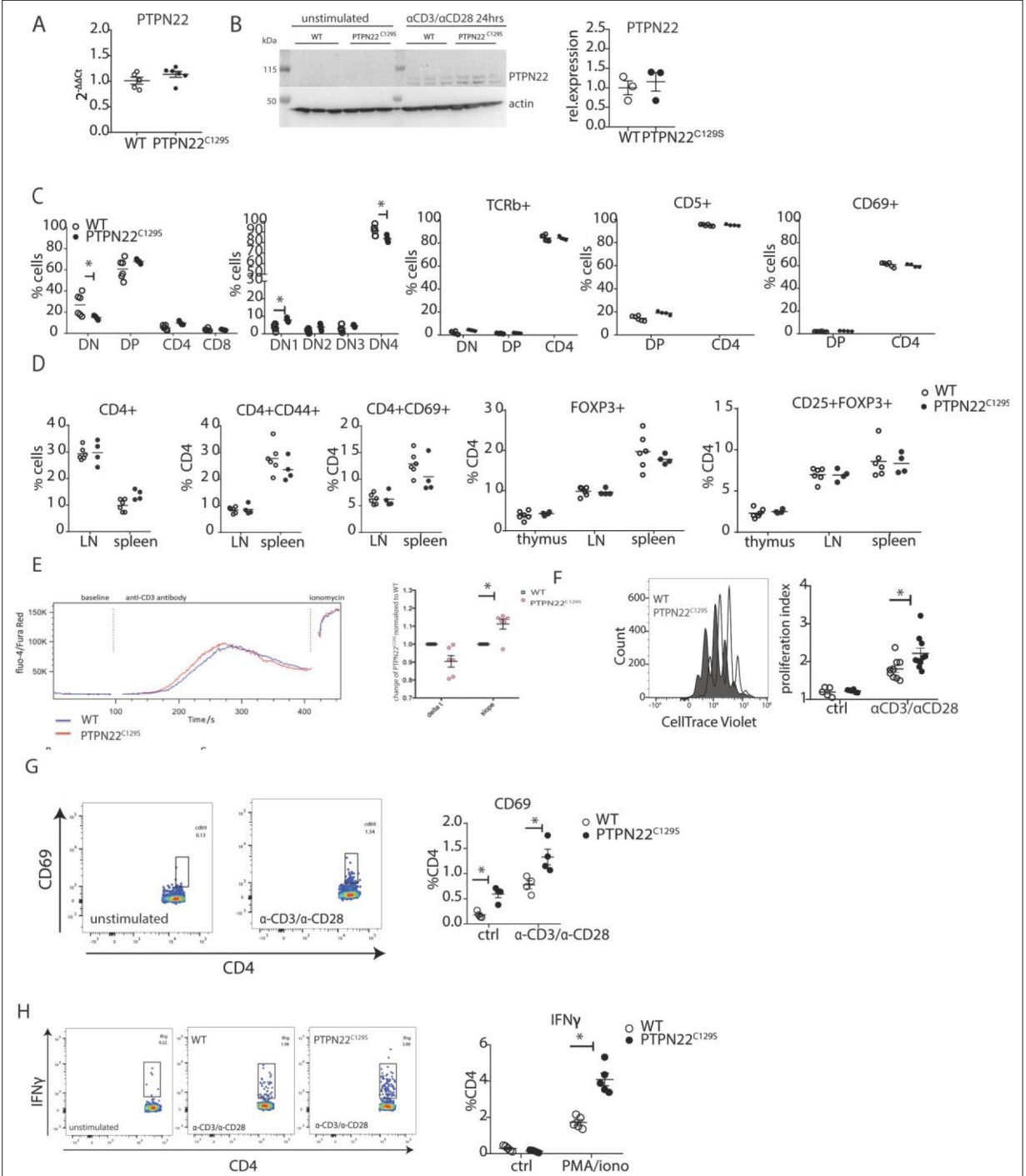

**Figure 5.** Enhanced activation and proliferation of PTPN22[C129S] T cells. (**A**) Gene expression of PTPN22 in splenocytes shown as fold change over WT. (**B**) Immunoblot analysis of PTPN22 expression in sorted CD4 T cells either unstimulated or activated in vitro with 1 µg/ml anti-CD3/CD28 for 24 hr. Each lane shows cells from a distinct mouse, and relative expression to the right was calculated by normalizing to loading control (actin). (**C, D**) Analysis of T cell populations in naïve mice via flow cytometry. (**C**) Analysis of double negative (DN), double positive (DP), and single positive CD4 and CD8 thymic populations in 10-week-old littermate mice as well as expression of TCRb, CD5, and CD69 on various subsets. (**D**) Frequencies of activated (CD44+/CD69+) and regulatory (FOXP3+) CD4+ T cells in thymus and secondary lymphoid organs. (**E**) Intracellular calcium measurement in CD4 T cells at baseline, after stimulation with anti-CD3, and ionomycin to achieve maximum $Ca^{2+}$ influx via staining with Fluo4 and FuraRed. Shown is a representative image of the change in ratio of fluo4 to FuraRed expression over time (blue: wild-type; red: PTPN22[C129S]). Quantification to the right shows slope value that describes how fast the peak of $Ca^{2+}$ influx is reached (*Bajnok et al., 2013*). (**F**) Proliferation of CD4+ T cells as assessed by CellTrace Violet dilution after 96 hr in vitro stimulation with 1 µg/ml anti-CD3/CD28. Proliferation Index is the total number of divisions divided by the number of cells

*Figure 5 continued on next page*

*Figure 5 continued*

that went into division. Representative proliferation peaks on the left (clear: wild-type; black: PTPN22[C129S]) and quantification on the right (ctrl refers to unstimulated samples). **(G, H)** Ex vivo-stimulated CD4 T cells were assessed for CD69 expression/IFNγ production after stimulation with anti-CD3/anti-CD28 or phorbol 12-myristate 13-acetate (PMA)/ionomycin, respectively. Representative gating shown. Error bars represent mean ± SEM.

The online version of this article includes the following source data and figure supplement(s) for figure 5:

**Source data 1.** Uncropped Western blot images showing PTPN22 expression in CD4+ T cells.

**Source data 2.** Raw images of PTPN22 expression in CD4+ T cells.

**Figure supplement 1.** Gene expression of PTPN22 in splenocytes from NCF1[m1J] and PTPN22[C129S]/NCF1[m1J] mice shown as fold change over NCF1[m1J].

**Figure supplement 2.** Percentage of B cells (B220+), T cells (CD3+), macrophages (CD11B+), and dendritic cells (CD11B+) in peripheral lymphoid organs as well as major histocompatibility complex (MHC) II expression on B cells, macrophages, and dendritic cells.

revealed differential expression of multiple genes under both untreated and BSO-treated conditions: upregulation of *Nudt16l1* (**Baciu et al., 2000**; **Denhez et al., 2002**), *Clic4* (**Suh et al., 2007**), *Mgmt* (**Boldogh et al., 1998**), *Hsp* (**Koliński et al., 2016**), *Serpina1e* (**Janciauskiene et al., 2017**), and *Fscn1* (**Liu et al., 2016**), as well as downregulation of *Fuca1* (**Vecchio et al., 2017**) and *Rps6kb1* (**Romanelli et al., 1999**), have all been shown to be associated with increased PKC signaling (**Figure 6D**).

Thus, PTPN22[C129S], which is prone to inactivation by oxidation and more resistant to activating reduction, with decreased catalytic activity, triggers enhanced T cell signaling. This suggests that redox regulation of PTPN22 is an important factor in control of inflammation, and that increased oxidation of PTPN22 has broad signaling effects that can yield aggravated inflammatory disease.

## Discussion

In this study, we investigated whether redox regulation of PTPN22 has an impact on inflammation and autoimmune arthritis. We found that the C129S mutation in *Ptpn22* leads to reduced catalytic activity and increased susceptibility to oxidative inactivation in vitro, causing concomitant upregulation of T cell activation and T-cell-mediated autoimmunity in a NOX2-dependent manner.

Redox regulation describes signaling effects that must be considered in relation to enzymatically driven oxidative and reductive pathways (**Sies and Jones, 2020**; **Arnér and Holmgren, 2000**). Significant ROS producers include the membrane-bound NADPH oxidase (NOX) complexes as well as the mitochondrial respiratory chain, modulating responses such as cell growth, death, and immune function under both physiological and pathological conditions (**Schieber and Chandel, 2014**; **Zhong et al., 2018**). The protective effects of ROS against autoimmune diseases in vivo are best evidenced in the *Ncf1[m1J]* mouse where diminished ROS leads to exacerbated inflammation in arthritis and lupus models (**Olofsson et al., 2003**; **Kelkka et al., 2014**). NOX2 is expressed in antigen-presenting cells and can upon antigen presentation expose T cells to hydrogen peroxide, that is, ROS (**Gelderman et al., 2006**; **Gelderman et al., 2007**). This ROS exposure regulates NFAT activation, IL2 production, and plasticity by both influencing the redox status of kinases involved and through metabolic reprogramming (**Kesarwani et al., 2013**; **Contento et al., 2010**; **Gelderman et al., 2007**). Reversible cysteine modifications have emerged as a major redox regulatory mechanism important for ERK1/2 phosphorylation, calcium flux, cell growth, and proliferation of naïve CD4+ and CD8+ T cells (**García-Santamarina et al., 2014**). Proteins containing cysteine residues, such as PTPs, are particularly susceptible to redox effects in part due to the low pKa value of their active site. Oxidation leads to inhibition of PTP activity, and several PTPs such as Src homology region 2 domain-containing phosphatase-1 and 2 (SHP1/2) and tyrosine-protein phosphatase non-receptor type 1 (PTP1B) are well studied in terms of their redox regulation (**Ostman et al., 2011**; **Michalek et al., 2007**; **Frijhoff et al., 2014**; **Tonks, 2003**). The thioredoxin system is important for its reductive PTP-activating capacity (**Dagnell et al., 2013**; **Dóka et al., 2020**; **Dagnell et al., 2017**).

PTPN22 is a PTP expressed in all hematopoietic cells and a known negative regulator of T cell signaling. The R620W variant of PTPN22 has been associated with an increased risk of several autoimmune diseases, amongst them diabetes, rheumatoid arthritis, and systemic lupus erythematosus. It has been shown to interfere with the binding of PTPN22 to the C-terminal SRC kinase (CSK) affecting downstream signaling. However, whether R620W is a gain-of-function or loss-of-function variant remains controversial (**Bottini and Peterson, 2014**; **Stanford and Bottini, 2014**; **Carmona**

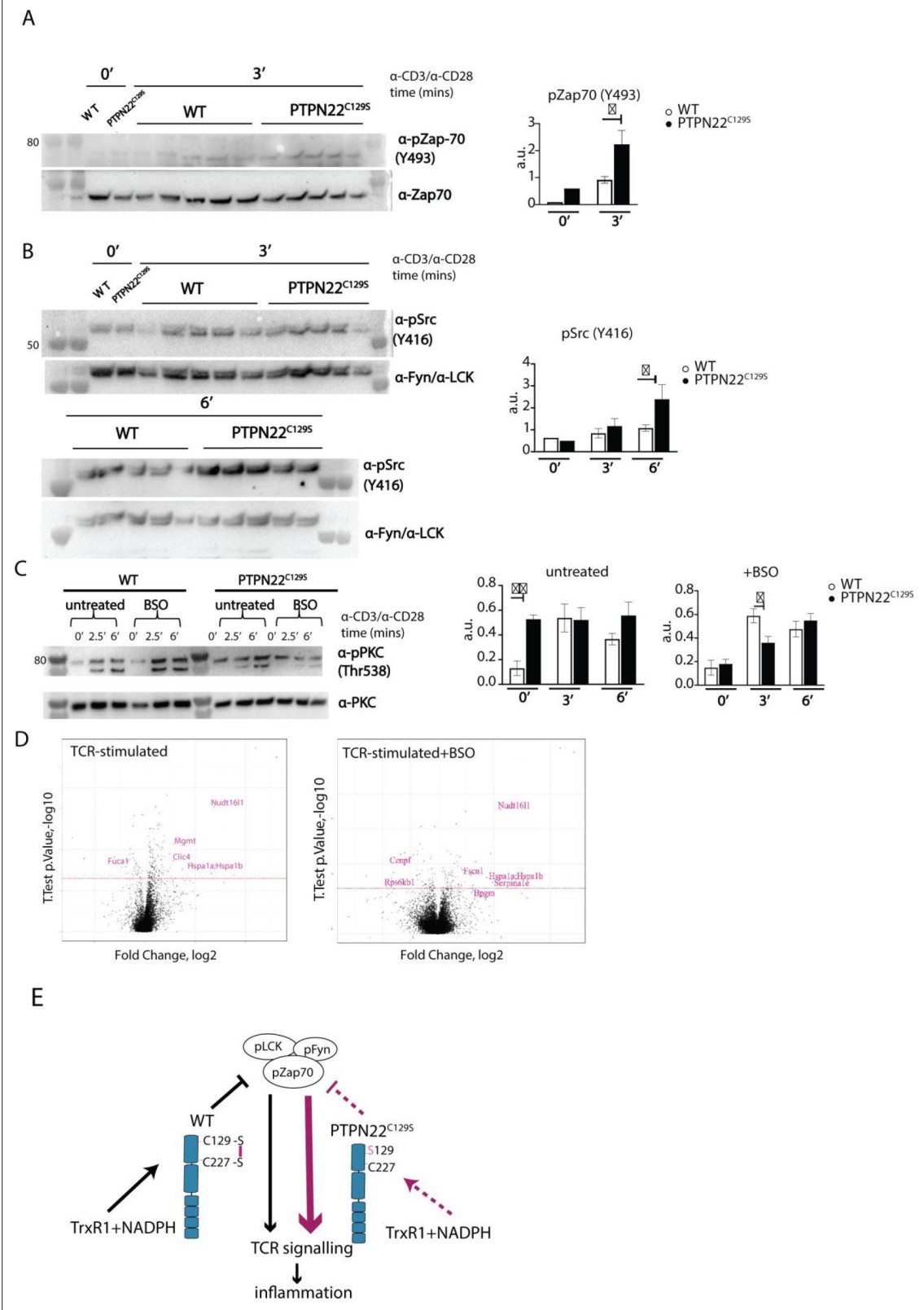

**Figure 6.** Enhanced phosphorylation of PTPN22^C129S targets and increased PKC θ expression. (**A, B**) Lymph node cells were stimulated for indicated time points with anti-CD3 plus anti-CD28, and phosphorylation was detected via immunoblotting of total cell lysates with antibodies against activated Zap-70 (**A**) and Src kinases (**B**). Antibody against p-Src (Y416) binds to Lck Y394 and Fyn Y417 with the top band representing pFyn and the bottom band pLCK, see *Davidson et al., 2016*. Each lane reflects distinct wild-type/PTPN22^C129S mice and unlabeled lanes show molecular weight markers.

*Figure 6 continued on next page*

*Figure 6 continued*

Quantification to the right was calculated as phosphorylation signal/total signal normalized to loading control. (**C**) Lymph node cells were cultured 24 hr with/without buthionine sulphoximine (BSO), a glutathione depletion agent, before in vitro activation with anti-CD3 plus anti-CD28 for indicated time points. Total lysate was analyzed for PKC $\theta$ expression via immunoblotting. Representative image shown. Quantification to the right was calculated as phosphorylation signal/total signal normalized to loading control (WT = 6, PTPN22$^{C129S}$ = 8). (**D**) Volcano plots comparing the proteomic profile of sorted CD4+ T cells from wild-type and PTPN22$^{C129S}$ mice, which were stimulated with anti-CD3 plus anti-CD28 for 5 hr either with or without prior treatment (24 hr) with BSO. Four mice per group were used for proteomic analysis, and p-values were calculated by Welch's *t*-test. Proteins that were significantly up/downregulated in PTPN22$^{C129S}$ mutants are highlighted in red. (**E**) A proposed model for redox regulation of PTPN22 in vivo: upon oxidative pressure within the cell, C227 of PTPN22 forms a disulfide bond with the back-door cysteine C129 as a means of protection from irreversible oxidation and inactivation. When the C227-C129 bond cannot be formed, as is the case in PTPN22$^{C129S}$ mutant, PTPN22 is less sensitive to reactivation by the thioredoxin system, which leads to increased T cell receptor (TCR) signaling and inflammation. Error bars represent mean ± SEM.

The online version of this article includes the following source data for figure 6:

**Source data 1.** Uncropped Western blot images showing phosphorylation of ZAP70, Src, and PKC upon CD3/CD28 stimulation.

**Source data 2.** Raw images of ZAP70, Fyn, and LCK expression.

**Source data 3.** Raw images of pZap70 and pSrc expression.

**Source data 4.** Raw images of pPKC expression.

**Source data 5.** Raw data from proteomics analysis.

*and Martín, 2018*). A possibility for redox regulation of PTPN22 was highlighted by Tsai et al., who discovered an atypical disulfide bond formation between the catalytic cysteine C227 and a 'back-door' cysteine C129, as visualized in a crystal structure of the catalytic domain of PTPN22 (*Tsai et al., 2009*). As physiologically relevant oxidative regulation of PTPN22 has thus far not been studied, we decided to mutate C129 into a serine, thereby abrogating the capacity of C129 to contribute to redox regulation of PTPN22, both in vitro and in vivo.

In our in vitro experiments, PTPN22$^{C129S}$ showed a lower basal catalytic activity, higher sensitivity to inhibition by oxidation, and a lower capacity for reactivation by the thioredoxin system compared to the wild-type enzyme. The reason behind the initial decrease in activity remains to be studied; the single cysteine-to-serine change may affect the PTPN22 protein conformation or dynamics and crystallography studies would be needed to address this. It should also be noted that the pure PTPN22$^{C129S}$ migrated slightly slower than wild-type PTPN22, despite the only difference between the two proteins being the single C129 residue. This also suggests an overall effect on protein characteristics by the C129S mutation, which should be studied further. Based on our findings, we propose a model for the redox regulation of PTPN22, with the disulfide bond that can be formed between C227 and C129 protecting the catalytic C227 residue from overoxidation to sulphinic or sulphonic acid. It is also possible that an alternative sulphenylamide motif can be made between the thiol group of the C227 with the peptide bond amine, which can also be reduced by the thioredoxin system, similar to that seen with PTP1B (*Dagnell et al., 2013*; *Schwertassek et al., 2014*) even if that motif was not seen in the crystal structure of PTPN22 (*Tsai et al., 2009*). This model explains how PTPN22$^{C129S}$ can still show activity and a certain protection against oxidation and reactivation by the thioredoxin system, but simultaneously be more susceptible to oxidation because the protective disulfide cannot be formed.

In T cells, the PTPN22$^{C129S}$ mutation led to enhanced T cell activation and proliferation. This was not due to altered PTPN22 expression levels, but rather a consequence of downstream signaling. The PTPN22 targets LCK, Fyn, and Zap70 showed increased phosphorylation, arguing for lower catalytic activity of PTPN22$^{C129S}$ in vivo and confirming our in vitro findings. In TCR signaling, phosphorylation of PKC-θ is a late-stage event that leads to the activation of transcription factors Nf-Kb, NFAT, and AP-1 controlling T cell functions down the line (*Wang et al., 2012*). We observed baseline upregulation of p-PKC-θ in PTPN22$^{C129S}$ T cells, which did not respond to further TCR stimulation. PKC phosphorylation is regulated by germinal center kinase-like kinase (GLK) and LCK (*Wang et al., 2012*; *Brezar et al., 2015*; *Gupta et al., 2008*). An increase in LCK phosphorylation as previously observed might therefore be causing increased p-PKC-θ. In an oxidative microenvironment created by BSO treatment, the initial difference in unstimulated cells disappeared and PTPN22$^{C129S}$ instead showed slower phosphorylation kinetics compared to wild-type. This might be mediated by direct effects of ROS on PKC as it contains several redox-sensitive residues that can lead to PKC inactivation (*Steinberg, 2015*). Proteomics data further corroborated the enhanced activation state. Interestingly, in

the *Ptpn22* mutant, we observed downregulation of CENPF, a cell cycle protein that was shown to be upregulated upon TrxR inhibition (*Selenius et al., 2012*).

Enhanced T cell activation led to increased inflammation in vivo, and we observed a pronounced skewing towards Th1 responses. A link between PTPN22 and Th1 responses has previously been established; PTPN22[-/-] and PTPN22[R619W] mice accumulate Th1 effector cells in their lymphoid organs with age or upon immune challenge. Th1 expansion in these mice was found to be dependent on LFA-ICAM1 interactions (*Sanchez-Blanco et al., 2018*). A collagen-induced arthritis (CIA) model produced no visible differences in inflammation in PTPN22[C129S] mice, in line with previous data showing that even a complete lack of PTPN22 does not result in significant differences in inflammation (*Figure 3—figure supplement 2*; *Sanchez-Blanco et al., 2018*). An explanation could be that the major pathogenic factor in the CIA model are autoreactive B cells producing pathogenic antibodies, rather than autoreactive T cells (*Holmdahl et al., 1990*).

In both T-cell-mediated inflammatory models (DTH and GPI) as well as on the Col2-T cell transgenic background, PTPN22[C129S] mice displayed higher T cell activity as well as more regulatory T cell markers such as FOXP3 and PD1 (shown for GPI model). A rise in Treg numbers is a measure to combat inflammation, but where they originate from is a matter of discussion. They can originate from the thymus and migrate to the tissue in response to inflammation or result from conversion of CD4+ CD25- naïve T cells in the periphery (*Yadav et al., 2013*). PTPN22 plays an important role in determining TCR signal strength during central tolerance; PTPN22-deficient mice show increased positive selection in the thymus as well as increased Treg numbers both in the thymus and periphery, leading to protection from the experimental autoimmune encephalomyelitis (EAE) model of autoimmunity (*Maine et al., 2012*; *Hasegawa et al., 2004*; *Fousteri et al., 2014*). In our study, Treg numbers were comparable between wild-type and PTPN22[C129S] thymi, suggesting peripheral induction of Tregs upon inflammation in PTPN22[C129S] mice. This may be a consequence of increased IFNɣ production by PTPN22[C129S] T cells as studies have also associated FOXP3 induction to Th1 responses; this mechanism links IFNɣ and IL27 with amplifying TGF-b-induced FOXP3 expression via STAT1 (*Ouaked et al., 2009*; *Littringer et al., 2018*). Previous studies have also shown the essential role of PKC-θ in NFAT-dependent FOXP3 expression. PKC-θ[−/−] mice have significantly reduced CD4+ FOXP3+ Tregs in the thymus, spleen, and lymph nodes (*Gupta et al., 2008*). Accordingly, the rise in Tregs in PTPN22[C129S] mice may be a consequence of the enhanced baseline PKC-θ signaling.

In summary, our results show how the activity of PTPN22, a gene where a risk variant is associated with autoimmune diseases, can be regulated by ROS through its noncatalytic C129 residue, which is likely to have a major impact on its function in addition to that of merely altered basal turnover (*Figure 6D*). It does not exclude additional modifications of other cysteines that have not been specifically addressed here. It has been notoriously difficult to target PTPs in vivo with drug therapies due to the nature of their active sites (*Stanford and Bottini, 2017*), but recent advances in targeting the redox status of T lymphocytes might represent a novel strategy to treat T-cell-driven diseases. Antibody-trapping of the oxidized form of PTP1B, a highly touted drug target for diabetes and obesity, has been shown to increase insulin signaling in vitro (*Stanford and Bottini, 2014*; *Krishnan et al., 2018*; *Stanford and Bottini, 2017*). It is therefore of importance to study the redox regulatory effects on the T cell machinery and its effects on immune processes. As such, our study contributes to the understanding of redox regulation of PTPN22 and possibly represents a new avenue for targeting PTPN22.

## Materials and methods
### Recombinant expression of PTPN22

The open-reading frame (ORF) of human PTPN22 catalytic domain (1–303 residues), codon optimized for recombinant expression in *Escherichia coli*, was synthesized by Integrated DNA Technologies, Inc. The ORF was subcloned into an in-house-developed pD441b plasmid (pD441b-HsPTPN22cd) that generates a fusion protein of His6-sfGFP-SUMO-HsPTPN22cd, where His6 is an N-terminal His-tag for IMAC purification, sfGPF (superfolder GFP; *Pédelacq et al., 2006*) included for enhanced protein folding, solubility and direct visibility, and small ubiquitin-related modifier (SUMO) being a 110-residue sequence recognized by SUMO protease ULP1 that hydrolyzes the peptide bond at the C-terminus of the SUMO domain, resulting in release of the target HsPTPN22cd from its N-terminal fusion

partner. The C129S mutation of human PTPN22 was introduced into the wild-type PTPN22-encoding plasmid pD441b-HsPTPN22cd by PCR using primer pairs HsPTPN22-C129S-fwd (GATCGGTCTCAT GGCGAGCATGGAGTACGAGATGGG) and HsPTPN22-C129S-rev (GATCGGTCTCCGCCATAACA ATGATCAGTAC), and the resulting plasmid was named pD441b-HsPTPN22cd-C129S. Both plasmids were respectively transformed into *E. coli* BL21 (DE3) strain for propagation and protein expression. The full sequences of the ORFs of the constructs for the wild-type PTPN22cd and its C129S mutant were verified by sequencing (Eurofins Biotech). Briefly, for each protein expression and purification, a 40 ml overnight culture was inoculated into 2 l terrific broth (TB) medium containing 50 μg/ml kanamycin in a 5-l bottle placed on a shaking incubator at 37°C. 4 hr after the inoculation, the incubator temperature was lowered to 25°C and 0.5 mM IPTG was added to induce protein expression overnight. The bacteria were harvested by centrifugation, suspended in IMAC binding buffer (50 mM Tris-HCl, 100 mM NaCl, 10 mM imidazole, pH 7.5), and lysed by sonication. The soluble fraction was recovered by centrifugation and purified by applying a HisPrep FF 16/10 column used with an ÄKTA explorer FPLC system (Cytiva Life Sciences). The eluted and purified fusion protein was subsequently treated with an in-house-expressed and purified His-tagged ULP1 (1% w/v), and the cleavage solution was again reapplied onto the HisPrep FF 16/10 column for separation of non-tagged target protein from its N-terminal His-tagged fusion partner as well as from the His-tagged ULP1. The target protein was then concentrated, buffer exchanged, and stored in the freezer. The purity of the product was greater than 95% as assessed by SDS-PAGE.

## PTP activity assay

PTP activity of recombinant wild-type PTPN22 and PTPN22$^{C129S}$ (1400 nM) was measured spectrophotometrically using as substrate 15 mM chromogenic *p*-nitrophenyl phosphate (pNPP) (P4744-1G, Sigma-Aldrich), as described previously (*Montalibet et al., 2005*). The absorbance increase rates were measured at 410 nm at 22°C using an Infinite M200 Pro plate reader (Tecan). Reduced PTPN22 (1400 nM, 49.9 μg/ml) was preincubated in 20 mM HEPES, 0.1 mM diethylenetriaminepentaacetic acid, and 100 mM NaCl buffer (pH 7.4) containing 0.05% BSA to prevent PTPN22 from time-dependent inactivation (*Dagnell et al., 2019*), together with 1 mM sodium azide. Sodium azide was used to inhibit any trace amounts of catalase. At the indicated concentrations, human Trx1, TrxR1, and NADPH (N7505-100MG, Sigma-Aldrich) were added (with Trx1 and TrxR1 expressed and purified as described in *Cheng and Arnér, 2017*). Variations in activity were observed between different batches of PTP purifications, and activities were thus always compared with the controls within each experiment.

## PTP treatment with $H_2O_2$ and bicarbonate

Pre-reduced and desalted PTPN22 was exposed to $H_2O_2$ and different components of the Trx system at the indicated time points followed by the addition of pNPP and measurement of activity. The activity after each $H_2O_2$ treatment was related to the activity of untreated PTPN22 incubated for the same time. For analysis of the effects of bicarbonate, similarly pre-reduced PTPN22 (1400 nM) was pre-incubated in 20 mM HEPES, 100 mM NaCl buffer, pH 7.4, containing 0.1 mM EDTA, 0.05% BSA. For treatment with oxidant, bicarbonate and $H_2O_2$ were premixed in the same buffer before addition to PTPN22 for treatment. Each PTPN22 in reaction mixture was exposed for the indicated times to $H_2O_2$ with or without bicarbonate (pH 7.4), and subsequent to this treatment, measurement of PTP activity was performed.

## PTP reactivation assay

PTPs from stock storage solutions were exchanged into reactivation buffer as described previously (*Parsons and Gates, 2013*) containing 0.5% (v/v) Tween 80. Subsequently, the PTPs were inactivated by treatment with 1 mM $H_2O_2$ for 5 min at 25°C. Following this inactivation incubation, 20 μg/ml catalase was added to quench excess $H_2O_2$ whereafter the reactivation experiment was performed by adding the components of the Trx system as indicated.

## Animals

### Establishment of mouse strains

#### Vector construction

A genomic fragment encompassing exons 2–10 of the *Ptpn22* gene was isolated from a BAC clone of C57BL/6 origin (no. RP23-189D14, Deutsches Ressourcenzentrum für Genomforschung). The TGT codon found in exon 5 of the *Ptpn22* gene and coding for the cysteine residue present at position 129 of PTPN22 was converted into a TCC codon coding for a serine. A *lox*P-tACE-CRE-PGK-gb2-*neor*-*lox*P cassette (NEO; *Mingueneau et al., 2008*) was introduced in the intron separating exons 5 and 6 of the *Ptpn22* gene, and a cassette coding for the diphtheria toxin fragment A abutted to the 5′ end of the targeting vector.

#### Isolation of recombinant ES clones

After electroporation of Bruce 4 C57BL/6 ES cells (*Köntgen et al., 1993*) and selection in G418, colonies were screened for homologous recombination by Southern blot. A probe specific to the NEO cassette was also used to ensure that adventitious nonhomologous recombination events had not occurred in the selected ES clones.

#### Production of mutant mice

Mice were handled in accordance with national and European laws for laboratory animal welfare and experimentation (EEC Council Directive 2010/63/EU, September 2010), and protocols approved by the Marseille Ethical Committee for Animal Experimentation. Mutant *Ptpn22*$^{C129S}$ ES cells were injected into FVB blastocysts. Germline transmission led to the self-excision of the NEO selection cassette in male germinal cells. Presence of the intended mutation was checked in homozygous *Ptpn22*$^{C129S}$ mice by DNA sequencing. Screening of mice for the presence of the *Ptpn22*$^{C129S}$ mutation was performed by PCR using the following oligonucleotides: 5′-CATGCAGGACTGTCCTCTCT-3′ and 5′- GTATTCTT GTCTCCCTTCCT-3′. This pair of primers amplifies a 283 bp band in the case of the wild-type allele and a 368 bp band in the case of the *Ptpn22*$^{C129S}$ allele. Mice harboring the *Ptpn22*$^{C129S}$ mutation have received the international strain designation C57BL/6-*Ptpn22*$^{tm1Ciphe}$, and founders were sent to Karolinska for further experiments.

The mice were backcrossed to C57Bl6/N.H2$^q$ (B6Q) mice. This genetic background was used in all experiments. In some experiments, we introgressed a TCR VDJ beta knock-in as well as a TCR alpha locus originally derived from the DBA/1 mouse (*Mori et al., 1992*). For other experiments, the Ncf1$^{m1j}$ mutation was inserted, leading to a functional impairment of the NOX2-mediated ROS response (*Sareila et al., 2013*). Mice were kept under specific pathogen-free (SPF) conditions in the animal house of the Section for Medical Inflammation Research, Karolinska Institute, Stockholm, Sweden. Animals were housed in individually ventilated cages containing wood shavings in a climate-controlled environment with a 14 hr light-dark cycle, fed with standard chow and water ad libitum. All mice were healthy and basic physiological parameters were not affected. All the experiments were performed with age- and sex-matched mice and in a blinded fashion. Experimental procedures were approved by the ethical committees in Stockholm, Sweden. Ethical permit numbers: 12923/18 and N134/13 (genotyping and serotyping), N35/16 (DTH, GPI, CIA).

## Delayed-type hypersensitivity

12-week-old mice were sensitized by intradermal injection of 100 µg rat collagen type II (Col2) in 100 µl of a 1:1 emulsion with complete Freund's adjuvant (BD, Difco, MI) and 10 mM acetic acid at the base of the tail. Day 8 after sensitization, the right ear was injected intradermally with 10 µl Col2 in PBS (1 mg/ml) whilst the control left ear was injected with 10 µl acetic acid in PBS. Ear swelling response was measured 0 hr, 24 hr, 48 hr, and 72 hr after challenge using a caliper. Change in ear thickness was calculated by subtracting the swelling of the PBS-injected ear from the swelling of the Col2-injected ear normalized to day 0 thickness. Ear tissue was processed according to *Mac-Daniel et al., 2016*.

## GPI-induced arthritis

Arthritis was induced by the hGPIc-c peptide (NH2-IWYINCFGCETHAML-OH; Biomatik) emulsified with an equal volume of complete Freund's adjuvant. Each mouse was intradermally injected with 100 µl emulsion (10 µg/mouse) at the base of the tail, and arthritis development was monitored using

a macroscopic scoring system where visibly inflamed ankles or wrists received five points each and inflamed toes/fingers were given one point.

## Collagen-induced arthritis

Arthritis was induced with 100 μg of heterologous rat Col2 in 100 μl of a 1:1 emulsion with CFA and 10 mM acetic acid injected intradermally at the base of the tail. Mice were challenged on day 28 with 50 μg of Col2 in 50 μl of IFA (BD, Difco) emulsion. Arthritis development was monitored using a macroscopic scoring system as described above.

## Cell culture

$10^6$ splenocytes or lymph node cells were cultured in 200 μl of complete DMEM per well in U-shaped bottom 96-well plates (Thermo) at 37°C and 5% $CO_2$. Complete DMEM consisted of DMEM + GlutaMAX (Gibco); 5% FBS (Gibco); 10 μM HEPES (Sigma); 50 μg/ml streptomycin sulfate (Sigma); 60 μg/ml penicillin C (Sigma); 50 μM β-mercaptoethanol (Gibco). FBS was heat-inactivated for 30 min at 56°C. The following stimuli were used: hGPIc-c (10 μM; Biomatik), mGPIc-c (10 μM, Biomatik), galCOL2259-273 (10 μg/ml, in-house), concanavalin A (ConA, 1 μg/ml), anti-mouse CD3 (1 μg/ml, 145-2C11, BD); anti-mouse CD28 (1 μg/ml, 37.51, BD), phorbol 12-myristate 13-acetate (20 ng/ml), and ionomycin (1 μg/ml).

## ELISA

Flat 96-well plates (Maxisorp, Nunc) were coated overnight at 4°C with 10 μg/ml rCol2/GPIp BSA in PBS. Plates were washed in PBS + 0.05% Tween and blocked with 1% BSA in PBS for 2 hr at 37°C and washed again. Mouse serum was added in dilutions varying from 1:100 to 1:10,000. Plates were incubated for 2 hr at 37°C, washed, and incubated with secondary antibodies (Southern Biotech: IgK [1170-05], IgG2b [1091-05]) for 1 hr at 37°C. After washing, 50 μl ABTS substrate (Roche) was added and signal was read at 405 nm (Synergy 2, BioTek).

## T cell proliferation

Proliferation of CD4+ T cells purified by negative sorting (Thermo, 11416D) was assessed using the CellTrace Proliferation Kit (Thermo, C34554) according to the manufacturer's instructions.

## ELISpot

EMD Millipore MultiScreen 96-Well Assay Plates were coated overnight at 4°C with the capture antibody in PBS. Coating solution was decanted and $10^6$ splenocytes or $5 \times 10^5$ lymph node cells were incubated 24 hr at 37°C together with various stimuli. After culture, plates were washed (0.01% PBS-Tween) and biotinylated detection antibodies were added to PBS. Plates were washed and Extra-vidin Alkaline phosphatase (Sigma) was added at a 1:2500 dilution in PBS (30 min, room temperature [RT]). Plates were washed before adding SIGMAFAST BCIP/NBT (Sigma) substrate solution. Once spots became visible, plates were washed in water and spots were counted using a CTL ImmunoSpot Analyzer. *Antibodies (Ab)*: IL-2 (capture Ab 5 μg/ml JES6-IA12; detection Ab 2 μg/ml biotinylated-JES6-5H4, in-house produced); IL-17A (capture Ab 5 μg/ml TC11-18H10.1; detection Ab 2.5 μg/ml TC11-8H4, BioLegend); IFNγ (capture Ab 5 μg/ml AN18; detection Ab 2.5 μg/ml biotinylated R46A2, in-house produced).

## qPCR

RNA was extracted from 1 to $2 \times 10^6$ cells using QIAGEN RNeasy columns according to the manufacturer's instructions. RNA concentration was determined using a NanoDrop 2000 (Thermo Scientific), and sample concentrations were normalized before proceeding with reverse transcription. cDNA synthesis was carried out using the iScript cDNA synthesis kit (Bio-Rad) according to the manufacturer's instructions. The qPCR reaction was carried out using the iQSYBR Green Mix (Bio-Rad) in white 96-well plates using a CFX96 real-time PCR detection system (Bio-Rad). *Actb* was used as an endogenous control. Data were analyzed according to the ΔΔCt method (*Livak and Schmittgen, 2001*). Primer sequences are listed in *Supplementary file 1*.

## Flow cytometry

Samples were stained with the indicated antibodies in 50 µl of PBS diluted 1:200 at 4°C for 20 min in the dark. Cells were washed once, fixed, and permeabilized for intracellular staining using BD Cytofix/Cytoperm (BD) according to the manufacturer's instructions. Cells were stained intracellularly with 50 µl of permeabilization buffer (BD) using the antibodies at a 1:100 final dilution for 20 min at RT. FoxP3 staining required nuclear permeabilization and was carried out using Bioscience Foxp3/Transcription Factor Staining Buffer. For intracellular cytokine staining, cells were stimulated in vitro with phorbol 12-myristate 13-acetate (PMA) 10 ng/ml, ionomycin 1 µg/ml, and BFA 10 µg/ml for 4–6 hr at 37°C prior to fixation, permeabilization, and staining. Antibodies are listed in *Supplementary file 2*.

## Ca$^{2+}$ flux

Lymph node and spleen cells were stained in prewarmed PBS + 1% FCS with Fluo4-AM (2 µM, Thermo) and FuraRed AM (4 µM, Thermo) at 37°C. Cells were washed in cold PBS + 1% FCS before staining for extracellular markers for 20 min. Baseline Ca$^{2+}$ flux was recorded for 100 s before 50 µl anti-CD3 (10 µg/ml; BD) stimulation was added; after 5 min, maximum flux was measured using ionomycin (1 µg/ml; BD). Relative calcium concentration was plotted as a ratio of Fluo3 to FuroRed emission using FlowJo.

## Protein isolation and SDS-PAGE

Total protein was isolated from 2 × 10$^6$ cells in 60 µl lysis buffer (M-PER, Thermo) with freshly added protease inhibitors (Halt cocktail 100x, Thermo). Lysates were centrifuged for 10 min at top speed, and protein concentrations of supernatants were measured using Pierce BCA Protein Assay Kit (Thermo, 23225). SDS-PAGE (4–12% NuPAGE Bis-Tris gel; Thermo Scientific) was run according to the manufacturer's instructions (45 min, 200 V, MOPS buffer).

## Western blot

Proteins were blotted onto a PVDF membrane (Millipore) for 1.5 hr at 35 V in NuPAGE transfer buffer (Thermo Fisher). Membranes were blocked for 1 hr at RT in blocking solution (0.05% PBS-Tween, 5% BSA). Incubation with primary antibodies (listed below) was performed overnight at 4°C in blocking solution. After incubation, membranes were washed in PBS-T and incubated with AffiniPure peroxidase-coupled goat anti-rabbit IgG(H+L) (final concentration 40 ng/ml, Jackson Laboratories) for 1 hr at RT. Membranes were washed and coated with ECL substrate solution (GE Healthcare) before imaging on a ChemiDoc XRS+ (Bio-Rad).

## Antibodies

Cell Signaling Technology: p-PKC (9377S), PKC (1364S), Zap70 (2705S), PTPN22 (14693S), p-Src (2101S), LCK (2752S), Fyn (4023S), H2B (12364S). Abcam: p-Zap70 (ab194800), actin (ab8227), and vinculin (ab129002).

## LC-MS sample preparation

For proteomics analysis, cells were collected after treatment, washed twice with PBS, and then lysed using 8 M urea, 1% SDS, and 50 mM Tris at pH 8.5 with protease inhibitors (Sigma; Cat# 05892791001). The cell lysates were subjected to 1 min sonication on ice using Branson probe sonicator and 3 s on/off pulses with a 30% amplitude. Protein concentration was then measured for each sample using a BCA Protein Assay Kit (Thermo; Cat# 23227). 6.8 µg of each sample was reduced with DTT (final concentration 10 mM) (Sigma; Cat# D0632) for 1 hr at RT. Afterward, iodoacetamide (IAA) (Sigma; Cat# I6125) was added to a final concentration of 50 mM. The samples were incubated at RT for 1 hr in the dark, with the reaction being stopped by addition of 10 mM DTT. After precipitation of proteins using methanol/chloroform, the semi-dry protein pellet was dissolved in 25 µl of 8M urea in 20 mM EPPS (pH 8.5) (Sigma; Cat# E9502) and was then diluted with EPPS buffer to reduce urea concentration to 4 M. Lysyl endopeptidase (LysC) (Wako; Cat# 125-05061) was added at a 1:75 w/w ratio to protein and incubated at RT overnight. After diluting urea to 1M, trypsin (Promega; Cat# V5111) was added at the ratio of 1:75 w/w and the samples were incubated for 6 hr at RT. Acetonitrile (Fisher Scientific; Cat# 1079-9704) was added to a final concentration of 20% v/v. TMTpro reagents (Thermo; Cat# 90110) were added 4× by weight to each sample, followed by incubation for 2 hr at

RT. The reaction was quenched by addition of 0.5% hydroxylamine (Thermo Fisher; Cat# 90115). Samples were combined, acidified by trifluoroacetic acid (TFA; Sigma; Cat# 302031 M), cleaned using Sep-Pak (Waters; Cat# WAT054960), and dried using a DNA 120 SpeedVac concentrator (Thermo). Samples were then resuspended in 20 mM ammonium hydroxide and separated into 96 fractions on an XBrigde BEH C18 2.1 × 150 mm column (Waters; Cat# 186003023) using a Dionex Ultimate 3000 2DLC system (Thermo Scientific) over a 48 min gradient of 1–63%B (B = 20 mM ammonium hydroxide in acetonitrile) in three steps (1–23.5%B in 42 min, 23.5–54%B in 4 min, and then 54–63%B in 2 min) at 200 µl/min flow. Fractions were then concatenated into 24 samples in sequential order (e.g., 1, 25, 49, 73). After drying and resuspension in 0.1% formic acid (FA) (Fisher Scientific), each fraction was analyzed with a 90 min gradient in random order.

## LC-MS analysis

Samples were loaded with buffer A (0.1% FA in water) onto a 50 cm EASY-Spray column (75 µm internal diameter, packed with PepMap C18, 2 µm beads, 100 Å pore size) connected to a nanoflow Dionex UltiMate 3000 UPLC system (Thermo) and eluted in an increasing organic solvent gradient from 4% to 28% (B: 98% ACN, 0.1% FA, 2% $H_2O$) at a flow rate of 300 nl/min. Mass spectra were acquired with an orbitrap Fusion Lumos mass spectrometer (Thermo) in the data-dependent mode with MS1 scan at 120,000 resolution and MS2 at 50,000 (@200 $m/z$) in the mass range from 400 to 1600 $m/z$. Peptide fragmentation was performed via higher-energy collision dissociation (HCD) with energy set at 35 NCE.

## Protein identification and quantification

The raw data from LC-MS were analyzed by MaxQuant, version 1.6.2.3 (*Cox and Mann, 2008*). The Andromeda engine (*Cox et al., 2011*) searched MS/MS data against UniProt complete proteome database (*Mus musculus*, version UP000000589, 22,137 entries). Cysteine carbamidomethylation was used as a fixed modification, while methionine oxidation and protein N-terminal acetylation were selected as a variable modification. Trypsin/P was selected as enzyme specificity. No more than two missed cleavages were allowed. A 1% false discovery rate was used as a filter at both protein and peptide levels. First search tolerance was 20 ppm (default), main search tolerance was 4.5 ppm (default), and the minimum peptide length was seven residues. After removing all the contaminants, only proteins with at least two unique peptides were included in the final dataset. Protein abundances were normalized by the total protein abundance in each sample in deep datasets. In the original dataset, protein abundances were normalized to ensure same median abundance across all channels in all replicates. Then for each protein, log2-transformed fold changes were calculated as a log2 ratio of the intensity to the median of all control replicates. All the proteins quantified in each experiment were used as the background.

## Statistical analysis

Statistical analysis was performed using GraphPad Prism v6.0. Statistical comparison of two unpaired groups was carried out using Mann–Whitney $U$ nonparametric test unless stated otherwise. Welch's variant of Student's $t$-test was used in indicated experiments. p-Values < 0.05 were considered statistically significant and are denoted as *p<0.05 or **p<0.01.

## Data availability

The mass spectrometry data that support the findings of this study have been deposited in ProteomeXchange Consortium (https://www.ebi.ac.uk/pride/) via the PRIDE partner repository (*Vizcaíno et al., 2014*) with the dataset identifier PXD025319.

## Acknowledgements

This work was supported by grants from the Knut and Alice Wallenberg foundation, the Swedish Medical Research Council, the Swedish Foundation for Strategic Research, the Hungarian Thematic Excellence Programme (TKP2020-NKA-26), and AstraZeneca. We thank F Fiore for the construction of the *Ptpn22*$^{C129S}$ mice. The work performed at Centre d'Immunophénomique was supported in part by the Investissement d'Avenir program PHENOMIN (ANR-10-INBS-07 to BM).

## Additional information

### Competing interests

Annika Åstrand, Rajneesh Malhotra: Is an employee of AstraZeneca. The author declares that no other competing interests exist. Bernard Malissen: Reviewing editor, eLife. The other authors declare that no competing interests exist.

### Funding

| Funder | Grant reference number | Author |
|---|---|---|
| Knut och Alice Wallenbergs Stiftelse | | Rikard Holmdahl |
| Swedish Medical Research Council | | Rikard Holmdahl |
| Swedish Foundation for Strategic Research | | Rikard Holmdahl |
| Hungarian Thematic Excellence Programme | | Elias Arnér |
| Investissement d'Avenir | | Bernard Malissen |

The funders had no role in study design, data collection and interpretation, or the decision to submit the work for publication.

### Author contributions

Jaime James, Data curation, Formal analysis, Investigation, Methodology, Writing - original draft, Writing – review and editing; Yifei Chen, Data curation, Formal analysis, Investigation; Clara M Hernandez, Amir A Saei, Hassan Gharibi, Data curation, Formal analysis, Investigation, Methodology; Florian Forster, Data curation, Florian Forster is affiliated with SCIOTEC Diagnostic Technologies GmbH. The author has no financial interests to declare., Formal analysis, Investigation, Methodology; Markus Dagnell, Data curation, Investigation, Methodology, Supervision; Qing Cheng, Data curation, Formal analysis, Investigation, Methodology, Supervision; Gonzalo Fernandez Lahore, Data curation, Investigation; Annika Åstrand, Conceptualization, Funding acquisition, Project administration, Resources, Supervision; Rajneesh Malhotra, Conceptualization, Funding acquisition, Project administration, Rajneesh Malhotra is affiliated with Sitryx. The author has no financial interests to declare., Supervision; Bernard Malissen, Conceptualization, Resources; Roman A Zubarev, Elias SJ Arnér, Methodology, Resources, Supervision, Writing – review and editing; Rikard Holmdahl, Conceptualization, Funding acquisition, Project administration, Resources, Supervision, Writing – review and editing

### Author ORCIDs

Jaime James http://orcid.org/0000-0003-3732-8838
Bernard Malissen http://orcid.org/0000-0003-1340-9342
Roman A Zubarev http://orcid.org/0000-0001-9839-2089
Rikard Holmdahl http://orcid.org/0000-0002-4969-2576

### Ethics

Experimental procedures were approved by the ethical committees in Stockholm, Sweden. Ethical permit numbers: 12923/18, N35/16 and N134/13 (genotyping and serotyping).

### Decision letter and Author response

Decision letter https://doi.org/10.7554/eLife.74549.sa1
Author response https://doi.org/10.7554/eLife.74549.sa2

## Additional files

### Supplementary files

- Supplementary file 1. Primers used in the study.
- Supplementary file 2. Antibody list.

• Transparent reporting form

## Data availability

All data generated or analysed during this study are included in the manuscript and supporting files. The mass spectrometry proteomics data have been deposited to the ProteomeXchange Consortium (http://proteomecentral.proteomexchange.org/cgi/GetDataset) via the PRIDE partner repository (Vizcaino et al., 2014) with the dataset identifier PXD025319.

The following dataset was generated:

| Author(s) | Year | Dataset title | Dataset URL | Database and Identifier |
|---|---|---|---|---|
| Amir AS, Roman AZ | 2021 | Redox regulation of PTPN22 affects the severity of T cell dependent autoimmune inflammation | http://www.ebi.ac.uk/pride/archive/projects/PXD025319 | PRIDE, PXD025319 |

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
