## [Editor Report]

This article documents a novel aspect of how T cell activation is regulated by the PTPN22 phosphatase, namely, reversible oxidation, which transiently reduces the activity of PTPN22 to allow the T cell antigen receptor to drive a strong activation signal. This compelling work adds to our understanding of how an immune response is initiated and provides new insights that could be exploited for the development of new drugs to treat immune-mediated diseases.

---

## [Decision Letter]

**Decision letter after peer review:**

Thank you for submitting your article "Redox regulation of PTPN22 affects the severity of T cell dependent autoimmune inflammation" for consideration by *eLife*. Your article has been reviewed by 3 peer reviewers, and the evaluation has been overseen by a Reviewing Editor and Betty Diamond as the Senior Editor. The following individual involved in review of your submission has agreed to reveal their identity: Tomas Mustelin (Reviewer #2).

Essential revisions:

1. Demonstration that the redox modifications of PTPN22 and PTPN22C129S occur in T cells.

2. Demonstration of the disulfide bond between C227 and C129 would be extremely valuable if possible, but it was appreciated by the reviewers that this might be too technically challenging, so is desired rather than essential.

*Reviewer #1 (Recommendations for the authors):*

The paper is very solid as it is. Nevertheless, I would recommend to the authors to invest some efforts in order to strengthen the physiological/clinical relevance of their findings.

*Reviewer #2 (Recommendations for the authors):*

This is an elegant study with ample controls, a clear logic, compelling data, and a model of PTPN22 regulation that fits well into the current understanding of PTP regulation and the role of PTPN22 in TCR signaling.

I have no concerns with this paper.

*Reviewer #3 (Recommendations for the authors):*

1 – In the Introduction, the sentence "Control of redox regulation of PTPs in cells depends upon the balance between inhibitory oxidation of the catalytic Cys residue and its activation by reduction, where the latter is typically maintained by the thioredoxin system." Should be tempered since glutathionylation of PTPs has also been reported.

2 – In the Introduction, the sentence "True redox regulation of PTPN22 requires that the fully oxidized and thus inactivated enzyme can be subsequently reactivated by reduction." Should be clarified. What is the meaning of "true oxidation"? Also, fully oxidized suggests that the enzyme should be terminally oxidized to SO3.

3 – Since the disulfide between C227 and C129 is based on a published crystal structure, the authors should provide an experimental proof of this disulfide, such as by MS analysis. See: DOI:https://doi.org/10.1074/jbc.M111899200

4 – Line 143, correct to C129S.

5 – There are some discrepancies between Fig1D,E, in which some protection is provided by Trx1 and Trp14, and Figure 2A, B. Can the authors explain why?

6 – Line 157, the authors should try to indicate that this is a hypothesis. Using language such as "This would disrupt the proposed mechanism…" would be less forceful.

7 – Reduced NOX2 activity in Figure 3C should be measured.

8 – PTPN22 oxidation should be assessed in Figure 4B.

9 – PTPN22 has been shown to be expressed in T cells before. In this context, results from 5B are difficult to interpret.

10 – Cells used in experiments should also be mentioned in the text since, CD4^+^ T cells are used for PTPN22 immunoblots, spleenocytes for qPCR, lymph node cells for Fyn/Lck and Zap immunoblots.

11 – The results from the BSO experiment need to be clarified.

12 – In 5 E and F, PEP is used for PTPN22. Please use PTPN22 throughout the manuscript.

13 – Paragraphs 2 and 3 of the discussion need to be tightened.

---

## [Author Response]

Essential revisions:1. Demonstration that the redox modifications of PTPN22 and PTPN22C129S occur in T cells.2. Demonstration of the disulfide bond between C227 and C129 would be extremely valuable if possible, but it was appreciated by the reviewers that this might be too technically challenging, so is desired rather than essential.

As we understand it, in both point 1 and 2 it is requested that the disulfide bond between C227 and C129, as previously suggested by Tsai et al., (2009) (1) with pure protein, should be documented to actually occur in the activated T cells. We fully agree that this would improve the study and we have therefore made several attempts to demonstrate this oxidation, or the oxidation state of the active site Cys residue in PTPN22 in situ. However, as we had also expected, it has proven to be technically very challenging (see attached the results of one of the experiments that have been performed). Nevertheless, as the functional consequence of the PTPN22 oxidation and the effect of the C129S mutation is clearly documented in the mouse, using in vivo experiments, we still think it is valid to conclude that the reversible oxidation state of PTPN22 as well as the involvement of the Cys129 residue regulates the function of PTPN22 in vivo, which is the main conclusion of our study.

Reviewer #2 (Recommendations for the authors):This is an elegant study with ample controls, a clear logic, compelling data, and a model of PTPN22 regulation that fits well into the current understanding of PTP regulation and the role of PTPN22 in TCR signaling.I have no concerns with this paper.

We thank the reviewer very much for the appreciative words and positive evaluation of the study.

Reviewer #3 (Recommendations for the authors):1 – In the Introduction, the sentence "Control of redox regulation of PTPs in cells depends upon the balance between inhibitory oxidation of the catalytic Cys residue and its activation by reduction, where the latter is typically maintained by the thioredoxin system." Should be tempered since glutathionylation of PTPs has also been reported.

Clearly glutathionylation can also be important, as could nitrosylation, formation of disulfides, sulfenylamide motifs, or oxidation to sulfinic or sulfonic acid. These potential modifications of the catalytic Cys residue are however all examples of inhibitory oxidation, so we believe that our original statement holds as written.

2 – In the Introduction, the sentence "True redox regulation of PTPN22 requires that the fully oxidized and thus inactivated enzyme can be subsequently reactivated by reduction." Should be clarified. What is the meaning of "true oxidation"? Also, fully oxidized suggests that the enzyme should be terminally oxidized to SO3.

We here emphasized that true redox regulation requires reversible oxidation/reduction steps, as otherwise complete oxidation would be irreversible and hardly considered a regulatory mechanism (rather an off-switch). Please note that we did not write ”true oxidation”. However, to avoid misunderstanding, we have now clarified the sentence as follows:

“True regulation of PTPN22 by redox mechanisms should require that the oxidized and thereby inactivated enzyme can be subsequently reactivated by reduction, as otherwise an oxidizing step could only represent an irreversible off-switch."

3 – Since the disulfide between C227 and C129 is based on a published crystal structure, the authors should provide an experimental proof of this disulfide, such as by MS analysis. See: DOI:https://doi.org/10.1074/jbc.M111899200

In the cited paper, MS analysis was performed on large quantity of oxidized PTEN (100 µg) obtained by incubating the purified protein for 5 min at room temperature. The identification of reduced and oxidized forms of PTEN in cells was performed by immunoblot analysis and not by mass spectrometry. Given the limitations of the MS technology, it currently doesn’t seem feasible to detect the disulfide between C227 and C129 with mass spectrometry in cellular lysate.

4 – Line 143, correct to C129S.

This has been corrected.

5 – There are some discrepancies between Fig1D,E, in which some protection is provided by Trx1 and Trp14, and Figure 2A, B. Can the authors explain why?

The experiments in Figure 1D and E show the protective activities of the Trx system proteins towards oxidation and thus inactivation of PTPN22 by addition of H2O2, while Figure 2A and B show the capacity of the Trx system proteins in reactivation of PTPN22 from a previous fully inactivated state. These activities are not comparable, as also discussed in the text.

6 – Line 157, the authors should try to indicate that this is a hypothesis. Using language such as "This would disrupt the proposed mechanism…" would be less forceful.

This has been addressed in the text.

7 – Reduced NOX2 activity in Figure 3C should be measured.

We admit that we do not directly show in this paper that the Ncf1m1j mutant mouse is deficient for NOX2 activity. However, this has been documented in numerous other papers and is established, and we have references to such papers. Therefore, we do not think it will add anything to show such data again. We also regularly test these mice for functional ROS activity, and they are also regularly screened for the Ncf1 mutation. We can of course show such data, if required, but it seem to us not meaningful.

8 – PTPN22 oxidation should be assessed in Figure 4B.

As mentioned above, demonstrating reversible redox modifications in T cells using our models has proven too technically challenging.

9 – PTPN22 has been shown to be expressed in T cells before. In this context, results from 5B are difficult to interpret.

Figure 5B serves to show that there is no difference in PTPN22 expression between wildtype and C129S CD4^+^ T cells. The lack of a band in unstimulated conditions is likely due to technical issues as it is well known that T cells have a low basal expression of PTPN22, but which increases upon T cell receptor stimulation. Additionally, though we chose to show expression after 24 hours, PTPN22 kinetics occur rapidly after TCR stimulation such as increased PTPN22 phosphorylation within 1-2 minutes (see Fiorillo et al., J Biol Chem, 2010; Yang et al., Sci. Signal.2020) (8) (9) as well as dephosphorylation of its targets in 2-5 minutes (Hasegawa et al., 2004) (10).

10 – Cells used in experiments should also be mentioned in the text since, CD4^+^ T cells are used for PTPN22 immunoblots, spleenocytes for qPCR, lymph node cells for Fyn/Lck and Zap immunoblots.

This information has now been added to the text.

11 – The results from the BSO experiment need to be clarified.

We agree that the exact mechanisms explaining the effects of BSO are difficult to know and the experiment could be better discussed. For once, our previous statement in the manuscript that said ” Treatment with BSO increases the oxidative burden in cells without directly affecting the thioredoxin system.” was an oversimplification and has now been clarified as follows: ” Treatment with BSO modulates the redox systems in cells by lowering the GSH levels and without directly affecting the thioredoxin system, at least initially before compensatory mechanisms may be activated.”. We have also revised the final sentence in the Results section from ” Thus, PTPN22C129S that is prone to inactivation by oxidation with decreased catalytic activity results in enhanced T cell signaling, which has broad signaling effects that can yield aggravated inflammatory disease.” to ” Thus, PTPN22C129S that is prone to inactivation by oxidation and more resistant to activating reduction, with decreased catalytic activity, triggers enhanced T cell signaling. This suggests that redox regulation of PTPN22 is an important factor in control of inflammation, and that increased oxidation of PTPN22 has broad signaling effects that can yield aggravated inflammatory disease.”

12 – In 5 E and F, PEP is used for PTPN22. Please use PTPN22 throughout the manuscript.

This has been changed.

13 – Paragraphs 2 and 3 of the discussion need to be tightened.

Yes, we agree. We have now reduced the length of these two paragraphs.

References:

1. Tsai SJ, Sen U, Zhao L, Greenleaf WB, Dasgupta J, Fiorillo E, et al. Crystal structure of the human lymphoid tyrosine phosphatase catalytic domain: insights into redox regulation. Biochemistry. 2009;48(22):4838-45.

2. Jackson SH, Devadas S, Kwon J, Pinto LA, Williams MS. T cells express a phagocyte-type NADPH oxidase that is activated after T cell receptor stimulation. Nat Immunol. 2004;5(8):818-27.

3. Gelderman KA, Hultqvist M, Holmberg J, Olofsson P, Holmdahl R. T cell surface redox levels determine T cell reactivity and arthritis susceptibility. Proc Natl Acad Sci U S A. 2006;103(34):12831-6.

4. Gelderman KA, Hultqvist M, Pizzolla A, Zhao M, Nandakumar KS, Mattsson R, et al. Macrophages suppress T cell responses and arthritis development in mice by producing reactive oxygen species. J Clin Invest. 2007;117(10):3020-8.

5. Dagnell M, Cheng Q, Rizvi SHM, Pace PE, Boivin B, Winterbourn CC, et al. Bicarbonate is essential for protein-tyrosine phosphatase 1B (PTP1B) oxidation and cellular signaling through EGF-triggered phosphorylation cascades. J Biol Chem. 2019;294(33):12330-8.

6. Gringhuis SI, Leow A, Papendrecht-Van Der Voort EA, Remans PH, Breedveld FC, Verweij CL. Displacement of linker for activation of T cells from the plasma membrane due to redox balance alterations results in hyporesponsiveness of synovial fluid T lymphocytes in rheumatoid arthritis. J Immunol. 2000;164(4):2170-9.

7. Carilho Torrao RB, Dias IH, Bennett SJ, Dunston CR, Griffiths HR. Healthy ageing and depletion of intracellular glutathione influences T cell membrane thioredoxin-1 levels and cytokine secretion. Chem Cent J. 2013;7(1):150.

8. Fiorillo E, Orru V, Stanford SM, Liu Y, Salek M, Rapini N, et al. Autoimmune-associated PTPN22 R620W variation reduces phosphorylation of lymphoid phosphatase on an inhibitory tyrosine residue. J Biol Chem. 2010;285(34):26506-18.

9. Yang S, Svensson MND, Harder NHO, Hsieh WC, Santelli E, Kiosses WB, et al. PTPN22 phosphorylation acts as a molecular rheostat for the inhibition of TCR signaling. Sci Signal. 2020;13(623).

10. Hasegawa K, Martin F, Huang G, Tumas D, Diehl L, Chan AC. PEST domain-enriched tyrosine phosphatase (PEP) regulation of effector/memory T cells. Science. 2004;303(5658):685-9.